# Learning Network Granger causality using Graph Prior Knowledge

**Lucas Zoroddu**                                                  *lucas.zoroddu@ens-paris-saclay.fr*
*Université Paris Saclay, Université Paris Cité, ENS Paris Saclay,*
*CNRS, SSA, INSERM, Centre Borelli, F-91190, Gif-sur-Yvette, France*

**Pierre Humbert**                                       *pierre.humbert@universite-paris-saclay.fr*
*Université Paris-Saclay, CNRS, Inria,*
*Laboratoire de mathématiques d'Orsay, F-91405, Orsay, France*

**Laurent Oudre**                                                  *laurent.oudre@ens-paris-saclay.fr*
*Université Paris Saclay, Université Paris Cité, ENS Paris Saclay,*
*CNRS, SSA, INSERM, Centre Borelli, F-91190, Gif-sur-Yvette, France*

**Reviewed on OpenReview:** *https://openreview.net/forum?id=DN6sut5fyR*

## Abstract

Understanding the relationships among multiple entities through Granger causality graphs within multivariate time series data is crucial across various domains, including economics, finance, neurosciences, and genetics. Despite its broad utility, accurately estimating Granger causality graphs in high-dimensional scenarios with few samples remains a persistent challenge. In response, this study introduces a novel model that leverages prior knowledge in the form of a noisy undirected graph to facilitate the learning of Granger causality graphs, while assuming sparsity. In this study we introduce an optimization problem, we propose to solve it with an alternative minimization approach and we proved the convergence of our fitting algorithm, highlighting its effectiveness. Furthermore, we present experimental results derived from both synthetic and real-world datasets. These results clearly illustrate the advantages of our proposed method over existing alternatives, particularly in situations where few samples are available. By incorporating prior knowledge and emphasizing sparsity, our approach offers a promising solution to the complex problem of estimating Granger causality graphs in high-dimensional, data-scarce environments.

## 1 Introduction

Multivariate time series analysis plays a crucial role in understanding and forecasting real-world phenomena involving multiple interdependent variables. Within such complex systems, Granger causality (Granger, 1969) graph has emerged as a powerful tool to uncover directional relationships and causal influences. It has therefore been the basis for a wide range of applications in fields such as economics (Stock & Watson, 2016), neuroscience (Seth et al., 2015), gene regulation (Yao et al., 2013), protein-protein interactions (Zou et al., 2010) etc. Granger causality assesses whether the past values of one variable can help predict the future values of another variable. For instance, we say that a random variable $X$ Granger-causes a variable $Y$ if the past values of $X$ contain information that enhances our ability to predict $Y$ beyond what we could achieve using only the historical values of $Y$ itself. While Granger causality is a powerful tool for studying relationships between two variables, many real-world phenomena involve multiple interconnected variables. Network Granger causality extends the basic Granger causality concept by assuming linear causal relationships between $p$ variables. Hence, it allows to capture the complex causal relationships within multiple variables. In this context, the aim is not to identify whether one variable influences another but to unravel the intricate causal structure that underlies an entire system. Network Granger causality is particularly

relevant in fields like neuroscience where researchers seek to understand how different regions of the brain interact and influence each other over time (Seth et al., 2015), in finance to capture market dynamics or in climatology to investigate complex interactions shaping Earth's climate system. It is also applicable to discover cause-and-effect relationships within genetic pathways and protein-protein interactions, shedding light on the regulatory mechanisms that govern these biological processes (Yao et al., 2013).

A conventional approach to learn Granger causality graphs involves estimating the parameters of a Vector AutoRegressive model (VAR) based on observed multivariate time series (Lütkepohl, 2005). However, inferring parameters of a VAR model can be non trivial, particularly when dealing with high-dimensional data and limited samples. To address this issue, researchers have studied various regularization techniques, approaching the problem from both frequentist and Bayesian perspectives. From a frequentist standpoint, Basu et al. (2015) explored the application of a Group Lasso penalty and its consistency properties. Since then, several sparsity-inducing penalties have been proposed in the works of Kock & Callot (2015), Lin & Michailidis (2017), and Nicholson et al. (2020). On the Bayesian side, diverse prior distributions governing the parameters of the VAR model were investigated in the literature. These include the use of Gaussian priors (Sims, 1993), Gaussian-inverted Wishart priors (Banbura et al., 2010), and hierarchical normal priors, as explored by Ghosh et al. (2018). However, as underlined by Duan et al. (2023), the resulting Granger causality graphs from these methods often exhibit undesired characteristics, ranging from excessive density to pronounced disconnection, potentially conflicting with established scientific knowledge.

To mitigate these issues and enhance the fidelity of Granger causality graphs, researchers have investigated for the incorporation of supplementary information in the form of network structures. For instance, in gene network analysis, genes are often grouped into distinct pathways, and it is a common observation that interactions within a pathway are more frequent than those bridging different pathways (Marlin et al., 2012). In response, Yao et al. (2013) have introduced penalization terms into the optimization process to ensure that the derived Granger graph aligns with this a priori knowledge. Another approach involves adopting a tree-rank prior distribution, enforcing the graph of Granger causalities to be a subgraph within the union of spanning trees Duan et al. (2023). More recently, Lin et al. (2024) proposed to solve a Structural Equation Model incorporating constraint to obtain a DAG. Furthermore, in scenarios characterized by signals from physical processes and recorded by sensors distributed across multiple spatial locations, the Euclidean k-NN (k-Nearest Neighbors) graph is often leveraged and finds widespread adoption within the domain of Graph Signal Processing (Ortega et al., 2018), where it serves as a foundational element for tasks such as signal filtering, denoising, and prediction. For instance, in (Isufi et al., 2019), a VAR(MA) model is fitted as a polynomial of the Laplacian matrix of the kNN graph. Nonetheless, most of works within the scientific literature operate under the idealized assumption of possessing a complete and precise prior graph — an unrealistic supposition in many real applications.

**Contributions:** In this paper, we propose an optimization problem for learning VAR model parameters by utilizing prior knowledge about relationships within time series data, represented in the form of an undirected graph (cf Figure 1). Our approach enables the incorporation of an incomplete and noisy prior undirected network when learning the Granger causality Network, and we jointly learn VAR parameters while denoising the initial graph. To do that, we propose a two-block coordinate descent algorithm and we prove that it converges to a set of stationary points, strengthening the reliability of our approach. Moreover, we show that the optimization problem corresponds to the computation of a Maximum A Posteriori (MAP) of a particular statistical model. To validate the effectiveness of our method, we conduct a series of experiments encompassing synthetic and real-world datasets. Our findings convincingly demonstrate the superiority of our proposed model over vanilla alternatives for varying noise levels over the prior network, particularly in scenarios marked by data scarcity. This underscores the potential of our method as a valuable tool for deriving Granger causality graphs in practical settings across a spectrum of applications.

## 2 Preliminaries

In this section we present the basics of (Network) Granger causality and its relationships with Vector Autoregressive (VAR) models. We also give a brief overview of existing methods for learning VAR parameters.

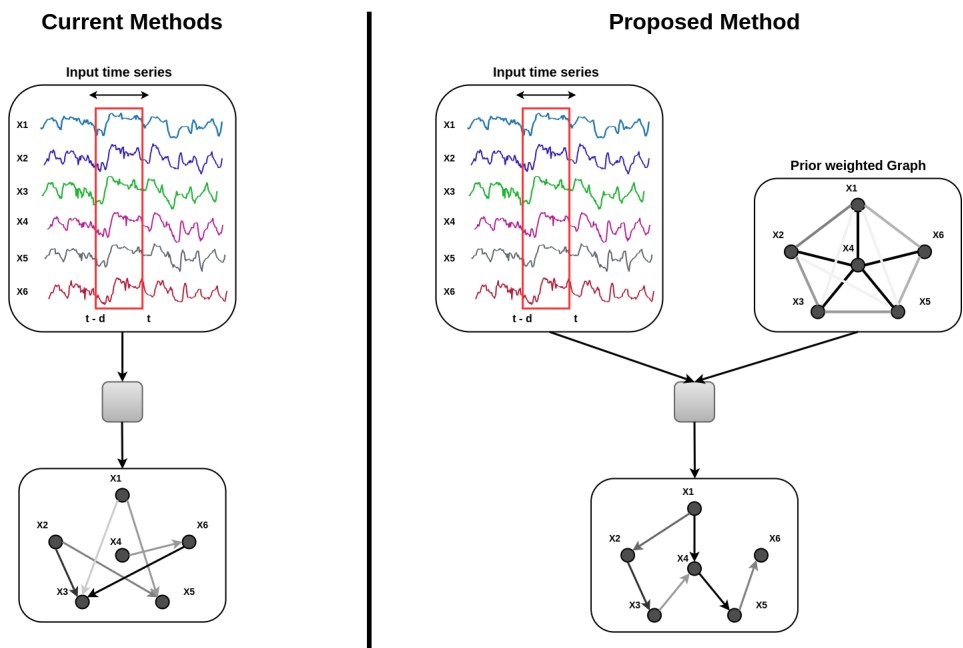

Figure 1: Diagram of the proposed method.
**Left:** Existing methods mainly infer causality graphs from time series only. **Right:** Our proposed method leverage both time series and a prior undirected graph (encoding 'likelihood' of link presence) to infer the causal graph.

## 2.1 Network Granger causality

In order to define the Network Granger causality, we recall the concept of classic Granger causality.

**Definition 1** (Granger causality (Granger, 1969)). *Let $x$ and $y$ be two stationary time series such that:*

$$y[t] = \sum_{i=1}^{d} \alpha_i y[t-i] + \sum_{i=1}^{d} \beta_i x[t-i] + \epsilon[t] , \tag{1}$$

*where $\epsilon[t] \sim \mathcal{N}(0, \sigma^2)$, $d$ is the order of the model, and the parameters $\alpha_i$, $\beta_i$ are fitted minimizing the least square error between $y$ and its reconstruction using* (1). *We say that $x$ Granger-causes $y$ if one of the $\{\beta_i\}_i$ is non zero.*

**Remark 2.** *In Granger (1969), the Granger causality is introduced as a statistical test based on the null hypothesis : $H_0 : \beta_1 = \beta_2 = ... = \beta_p = 0$. One can use a F-test to know whether the null hypothesis if rejected or not.*

**Remark 3** (Link between causality and Granger causality). *While Granger causality can effectively capture causal relationships, its interpretation leans more towards predictive power than a direct expression of true causality. For instance, if $x$ and $y$ are driven by an unknown $z$, it is possible that $(x, y)$ rejects the null hypothesis whereas there is no causality between them.*

To generalize the Granger causality to $p$ variables, we introduce Vector Autoregressive (VAR) models. A VAR model of order $d \geq 1$, denoted VAR($d$), explains the values of a multivariate time series at time $t$ using a linear combination of its $d$ previously observed values.

**Definition 4** (Vector Autoregressive Model (VAR)). *Given $d$ matrices $(\mathbf{C}^\tau)_{\tau=1}^d$ in $\mathbb{R}^{p \times p}$, a VAR(d) is defined at each time $t = 1, 2, \ldots$ by:*

$$X[t] = \sum_{\tau=1}^{d} \mathbf{C}^\tau X[t-\tau] + \varepsilon[t] , \tag{2}$$

*where $X[\cdot] = (X_1[\cdot], ..., X_p[\cdot])$ is a random p-dimensional time series and $\varepsilon[t] \sim \mathcal{N}(0, \sigma^2 I_p)$, $\sigma > 0$, is some innovation noise.*

Traditionally, Granger causality has been based on the assumption of a VAR model Lütkepohl (2005), primarily conducting tests on the VAR coefficients in a bivariate context. However, in complex real-world systems with numerous time series, analyzing the relationship between only two series can result in misleading conclusions due to confounding factors Lütkepohl (2005). In practice, VAR models are often used to analyze relationships between several variables of interest. Indeed, the matrices $\{\mathbf{C}^\tau\}_{\tau=1}^d$ in Equation (2) capture specific temporal dependencies between the $p$ dimensions and are associated with the notion of Network Granger causality (NGC), which was then introduced as a generalization of the Granger causality in Basu et al. (2015) to adjust for possible confounders or jointly consider multiple series.

**Definition 5** (Network Granger causality (Basu et al., 2015))**.** *Let $X[t] = (X_1[t], ..., X_p[t]) \in \mathbb{R}^p$ for $t = 1, 2, \ldots$, be a p-dimensional time series following the VAR model defined in Eq. (2). Then, $X_j[\cdot]$ is called a Granger cause of $X_i[\cdot]$ if at least one matrix entry $\{\mathbf{C}_{i,j}^\tau , \ \tau = 1, ..., d\}$ is non-zero.*

The intuition is that if $\mathbf{C}_{i,j}^\tau \neq 0$ for some $\tau$, like in the bivariate case, it means that past values of $X_j$ are useful to predict future values of $X_i$, independently from the other variables.
Indeed, we can reduce the model to a bivariate case for each couple $(i, j)$ writing:

$$X_i[t] = \sum_{\tau=1}^d \sum_{\substack{1 \leq k \leq p \\ k \neq j}} \mathbf{C}_{i,k}^\tau X_k[t - \tau] + \sum_{\tau=1}^d \mathbf{C}_{i,j} X_j[t - \tau],$$

and then use the definition of the classical Granger causality.

Note that, like for the classical Granger causality ($p = 2$), NGC does not necessarily capture true causal relationships, but rather indicates the power of prediction of some variables to others. Nevertheless, NGC remains a powerful tool for understanding interactions between random time series, and its estimation is of practical interest.

## 2.2 Estimation of the VAR parameters

Given $N$ samples $X^{(n)}[t] \in \mathbb{R}^p$ , $n = 1, ..., N$, at each time $t \in [\![1, d]\!] = $ stored in $\mathbf{X}[t] \in \mathbb{R}^{p \times N}$, one possibility to estimate the VAR parameters $\{\mathbf{C}^\tau\}_{\tau=1}^d$ is to maximize the likelihood of Model (2). This leads to the following Least Square problem:

$$\hat{\mathbf{C}}^{1:d} = \arg \min_{\mathbf{C}^1, ..., \mathbf{C}^d} \ \frac{1}{2N} \left\| \mathbf{X}[t] - \sum_{\tau=1}^d \mathbf{C}^\tau \mathbf{X}[t - \tau] \right\|_F^2 . \tag{3}$$

This minimisation can be divided into $p$ independent sub-problems, which are easier to handle. For $j = 1, ..., p$, they are given by:

$$\arg \min_{\mathbf{C}_{j:}^1, ..., \mathbf{C}_{j:}^d} \ \frac{1}{2N} \left\| \mathbf{X}_j[t] - \sum_{\tau=1}^d \mathbf{C}_{j:}^\tau \mathbf{X}[t - \tau] \right\|_2^2, \tag{4}$$

where $\mathbf{C}_{j:}^\tau$ refers to the $j$-th row of $\mathbf{C}^\tau$. Each sub-problem (4) is a standard linear problem and can therefore be solved efficiently.

In high dimensional settings, it can still be challenging to fit VAR($d$) models because the model has $d \times p^2$ degrees of freedom. Some regularization approaches were developed to overcome this limitation, both from a frequentist and Bayesian point of view. Several authors suggest adding a penalty term to learn the VAR parameters. Thus, the optimisation problem becomes:

$$\arg \min_{\mathbf{C}_{j:}^1, ..., \mathbf{C}_{j:}^d} \ \frac{1}{2N} \left\| \mathbf{X}_j[t] - \sum_{\tau=1}^d \mathbf{C}_{j:}^\tau \mathbf{X}[t - \tau] \right\|_2^2 + \lambda \mathcal{P}_\tau(\mathbf{C}_{j:}^1, ..., \mathbf{C}_{j:}^\tau) , \tag{5}$$

| Method | Penalization term | Guarantees | Statistical Model |
|---|---|---|---|
| Ridge | $\lambda\|u\|_2^2$ | – | $u \sim \mathcal{N}(0, I_p)$ |
| Lasso | $\lambda\|u\|_1$ | Selection consistency | $u_i \overset{iid}{\sim} Laplace(0, 1)$ |
| Adaptive Lasso | $\lambda \sum_k w_k |u_k|$ | Selection consistency (Zou, 2006) | $u_i \overset{ind}{\sim} Laplace(0, \frac{1}{w_i})$ |
| Group Lasso | $\lambda \sum_g w_g \|u_{[g]}\|_2$ | Direction consistency (Basu et al., 2015) | $u_g | \tau_g^2, \sigma^2 \overset{ind}{\sim} \mathcal{N}\left(\mu_g, \tau_g^2 \sigma_g^2 I_{m_g}\right)$, |
| | | | $\tau_g^2 \sim \text{Gamma}\left(\frac{m_g+1}{2}, \frac{\lambda^2}{2}\right)$ |

Table 1: Main regularizations for linear regression.

where $(\mathcal{P}_\tau)_{\tau=1}^d$ are some penalization terms.

The Lasso shrinkage technique (L1 penalization), as introduced by Tibshirani ((Tibshirani, 1996)), has been integrated into various models to account for specific data properties. In the study by Basu et al. ((Basu et al., 2015)), for example, the Group Lasso is proposed and explored to leverage knowledge of grouping structures. Lasso shrinkage proves effective in enhancing lag order selection, as demonstrated in the work of Shojaie and Michailidis ((Shojaie & Michailidis, 2010)) using a truncated Lasso estimator and in Nicholson et al.'s ((Nicholson et al., 2020)) hierarchical model.

The Adaptive Lasso (or weighted Lasso), introduced by Zou ((Zou, 2006)), finds applications in improving variable selection ((Zou, 2006)), addressing non-stationary time series, and proposing online algorithms ((Messner & Pinson, 2019)) by leveraging previous parameters to estimate the next ones. It's worth noting that the conventional approach for utilizing Adaptive Lasso involves initially solving the least square problem (without constraints) to obtain an initial estimator $\widehat{\mathbf{C}}_{OLS}$, followed by setting the weights to $wi, j = \frac{1}{|\widehat{\mathbf{C}}_{i,j}|}$.

Additionally, Adaptive Lasso has proven to be a relevant method for incorporating prior knowledge about parameters, as exemplified by the consideration of past values in ((Messner & Pinson, 2019)). Another commonly used estimator, the Ridge estimator, is also examined in Giovanni's study ((Ballarin, 2022)) within the context of VAR models, where the authors discuss the implications of anisotropic penalization. It's important to note that for some of these models, theoretical guarantees have been established regarding the selection consistency of Lasso-based methods (refer to Table 1).

As highlighted in the introduction, some Bayesian models have been introduced to learn VAR parameters by incorporating prior distributions. These priors aim to prioritize certain lags, with lower lags commonly assumed to be more informative than higher ones, or to impose specific structures on the graph, such as a spanning trees structure ((Sims, 1993; Banbura et al., 2010; Ghosh et al., 2018; Duan et al., 2023)). A comprehensive survey of existing Bayesian VAR models and their associated sampling algorithms is presented in the work of Miranda Agrippino and Ricco ((Miranda Agrippino & Ricco, 2018)).

However, note that in some cases, frequentist penalizations correspond to compute the Maximum Likelihood Estimator (MLE) of a particular Bayesian model, so it can be useful to consider a model from both point of view to have a deeper understanding of the solution. We refer to Table 1 for the correspondences between conventional penalization terms and their associated Bayesian models.

## 3 Model and Framework

In this section, we present our main contributions. In Section 3.1, we first introduce an optimization problem allowing to incorporate graph prior knowledge in the estimation of the parameters of a VAR model. Then, in Section 3.4, we describe an algorithm based on alternating minimization to solve this problem. In section 3.5, we prove that this algorithm converges to a stationary point and that its time complexity for a fixed number of iterations is of the same order as a Lasso estimator. Finally, we link this optimization problem to a maximum a posteriori of a certain statistical model, which is presented in Section 3.6.

In the following, we make the two classical assumptions:

**Assumption 1.** *The time series are generated by a* $\mathrm{VAR}(1)$ *model:*

$$X[t] = \mathbf{C}X[t-1] + \varepsilon[t] \,, \tag{6}$$

*so we only need to learn one matrix* $\mathbf{C}$. *However, note that the assumption* $d = 1$ *is not limiting since a* $\mathrm{VAR}(d)$ *model can always be written as a* $\mathrm{VAR}(1)$ *model (Lütkepohl, 2005). This generalization is detailed in Section 3.7.*

**Assumption 2.** *Hamilton (1994) showed that obtaining reliable estimates for* $\mathrm{VAR}$ *parameters necessitates stationary of* $X[\cdot]$. *Subsequently, in Lütkepohl (2005)'s work, it was established that the model* (6) *is stable if and only if* $\rho(\mathbf{C}) < 1$, *where* $\rho$ *is the spectral radius. This assumption will be maintained in our analysis.*

### 3.1 Problem formulation

In general, the estimation of the parameters of the VAR model (2), i.e. the matrix $\mathbf{C}$, requires the observation of a long stationary realization of the $p$-dimensional time series. However, in many applications, we observe only short replicas of the time series, and additional information must be incorporated into the model to obtain accurate estimates.

We propose to leverage prior knowledge on the structure of the matrix $\mathbf{C}$ by assuming that it is statistically related to a given matrix $\mathbf{A}^{\mathrm{prior}}$ which is the adjacency matrix of an undirected graph encoding a priori relationships between the individual dimensions. More specifically, the idea is to encode the following property: if two nodes $(i,j)$ are not likely to be linked, $\mathbf{A}^{\mathrm{prior}}_{i,j}$ is small and the value of the associated coefficient $\mathbf{C}_{i,j}$ is more likely to be closed to zero (meaning that there is no Granger causality between the two time series $X_i[\cdot]$ and $X_j[\cdot]$). While forbidding certain links in causal discovery is a standard approach and it used in recent methods like PCMCI Runge et al. (2019), this assumption may be too strong and this issue will be addressed in the paper (cf Remark 8). In addition, since the prior information is rarely perfectly accurate for most applications, we assume that $\mathbf{A}^{\mathrm{prior}}$ is not necessarily the optimal prior knowledge. So we want to allow the model to refine this prior through iterations, i.e. computing a matrix $\mathbf{A}$ close to $\mathbf{A}^{\mathrm{prior}}$ but not equal. To understand the relationships between these matrices, a statistical point of view is derived in 3.6 and provides the mathematical relationships between $\mathbf{C}, \mathbf{A}^{\mathrm{prior}}$ and $\mathbf{A}$.

Formally, let consider $N$ independent wide-sense stationary multivariate time series (Assumption 2) $X^{(1)}[1:d],...,X^{(N)}[1:d]$. In addition, suppose that we have access to a matrix $\mathbf{A}^{\mathrm{prior}}$, summarizing our prior knowledge on the relationships between each pair of variables. To estimate the VAR parameters, we introduce the following optimization problem:

$$\min_{\mathbf{A}\,,\,\mathbf{C}} \ \frac{1}{N} \sum_{n=1}^{N} \left\| X^{(n)}[t] - \mathbf{C}X^{(n)}[t-1] \right\|_2^2 + \lambda \sum_{1 \le i < j \le p} \frac{|\mathbf{C}_{i,j}| + |\mathbf{C}_{j,i}|}{\mathbf{A}_{i,j}}$$
$$+ 2\lambda \sum_{1 \le i < j \le p} \log(2\mathbf{A}_{i,j}) + \gamma \left\| \mathbf{A} - \mathbf{A}^{\mathrm{prior}} \right\|_F^2 \,, \tag{7}$$
$$\text{subject to} \ \ \mathbf{A}_{i,j} \ge 0 \,, \ \mathbf{A}_{i,j} = \mathbf{A}_{j,i} \,, \ \ 1 \le i < j \le p \,.$$

Equation (7) contains 3 terms:

(i) $\frac{1}{N} \sum_{n=1}^{N} \left\| X^{(n)}[t] - \mathbf{C}X^{(n)}[t-1] \right\|_2^2$ corresponds to the Least Square problem objective: this term allows to measure the difference between the original signals and their reconstructions. Recall that we only consider in this section $\mathrm{VAR}(1)$ model, hence $t = 2$.

(ii) $\sum_{1 \le i < j \le p} \frac{|\mathbf{C}_{i,j}| + |\mathbf{C}_{j,i}|}{\mathbf{A}_{i,j}} + 2 \sum_{1 \le i < j \le p} \log(2\mathbf{A}_{i,j})$ is the penalization term that takes into account the graph prior knowledge. This penalization is inspired by the one used in Adaptive Lasso models, where the terms $1/\mathbf{A}_{i,j}$ act as weights. The higher the $\mathbf{A}_{i,j}$, the closer $i$ and $j$ are in the graph, and the lower the penalty for $\mathbf{C}_{i,j}$ and $\mathbf{C}_{j,i}$. It should be noted that the additional term composed of the sum of log is a normalization term linked to the associated statistical model (see Section 3.6).

(iii) $\|\mathbf{A} - \mathbf{A}^{\mathrm{prior}}\|_F^2$ is a regularization term to take into account that $\mathbf{A}^{\mathrm{prior}}$ is not necessarily the optimal prior knowledge and could therefore be further refined. Actually, adding this term in the optimization problem is equivalent to imposing a normal prior distribution to the coefficient of $\mathbf{A}_{i,j}$ (see Section 3.6). In practice, this term allows to increases the robustness to the prior knowledge noise.

The symmetry constraint $\mathbf{A}_{i,j} = \mathbf{A}_{j,i}$ for $1 \leq i < j \leq p$ is applied as $\mathbf{A}$ represents the adjacency matrix of the denoised undirected prior graph knowledge.

**Remark 6.** *This work presents a model leveraging a prior undirected weighted graph. This model choice is based on the idea that for some applications, we have ideas about which variables are linked together but we do not know the directions of causality ($i \mapsto j$ or $j \mapsto i$?). However, it is possible to slightly modify the model to leverage a directed graph prior knowledge (the optimization problem remains the same except that we split the part with $\mathbf{A}_{i,j}$ and that we remove the symmetry constraint).*

Finally, the problem (7) depends on two hyper parameters $\lambda$ and $\gamma$. $\lambda$ controls the sparsity of the learned graph and $\gamma$ controls the confidence in the prior graph.

### 3.1.1 Interpretation of the optimization problem

- $\mathbf{A}^{\mathrm{prior}}$ is a $p \times p$ symmetric matrix, seen as the adjacency matrix of an undirected graph, corresponding to the prior knowledge of the Network Granger causality structure.

- $\mathbf{C}$ is a $p \times p$ matrix corresponding to the VAR(1) parameters and seen as the adjacency matrix of the directed graph of Granger causality.

- $\mathbf{A}$ is a $p \times p$ symmetric matrix, seen as the adjacency matrix of an undirected graph, corresponding to the denoised prior graph $\mathbf{A}^{\mathrm{prior}}$ during the optimization process.

- $X^{(i)}[t]$ ($t = 1, 2$) are the multivariate time series in $\mathbb{R}^p$ generated by the VAR(1) model with parameters $\mathbf{C}$.

The idea of this optimization problem is to find a sparse matrix $\mathbf{C}$ allowing to forecast $X$ using $\mathbf{A}^{\mathrm{prior}}$ as a first estimation of the parameters model. The main advantage here is that the prior network is taking into account during the learning process and allows to perform a relevant variable selection. Indeed, even though the variable selection consistency is proved for estimator like Lasso or Adaptive Lasso (cf (Zou, 2006)), this property holds for an infinite number of samples. Here, by assigning weights to pairwise relationships, our method allows to guide the variable selection using (noisy) prior knowledge in order to be efficient in settings with few samples available.

## 3.2 Motivating examples

Before entering in the details of the solving method, we illustrate here some use cases for which our method could be used, and how the graph $\mathbf{A}^{\mathrm{prior}}$ can be constructed depending on the problem.

**Multi-sensor networks:** In the context of multi-sensor networks, $\mathbf{A}^{\mathrm{prior}}$ can be computed by focusing on the distances between the sensors. For certain applications, the recordings are associated with physical processes, suggesting that causal links are more likely to exist between nearby sensors than distant ones: for instance, temperature data are governed by complex equations but exhibit spatial continuity. Consequently, in the experiment with the Molene dataset (Sec. 4.5.2), $\mathbf{A}^{\mathrm{prior}}$ is computed applying a Gaussian kernel to the pairwise distances: $\mathbf{A}_{i,j}^{\mathrm{prior}} = K_\sigma(d(x_i, x_j))$ in order to increase the penalization of causal links between far away sensors (cf Figure 2).

**Protein interactions:** Expert knowledge from the literature can be used to construct a score that reflects how likely two proteins interact based on previous experiments. In this case, $\mathbf{A}_{i,j}^{\mathrm{prior}}$ is the score of the couple (i,j). This can lead to prior knowledge graph about pairwise proteins interaction allowing to lead the causal discovery process. An interesting example is the one presented in Carlin et al. (2017) by aggregating various established pathways from Pathway Common Cerami et al. (2010)

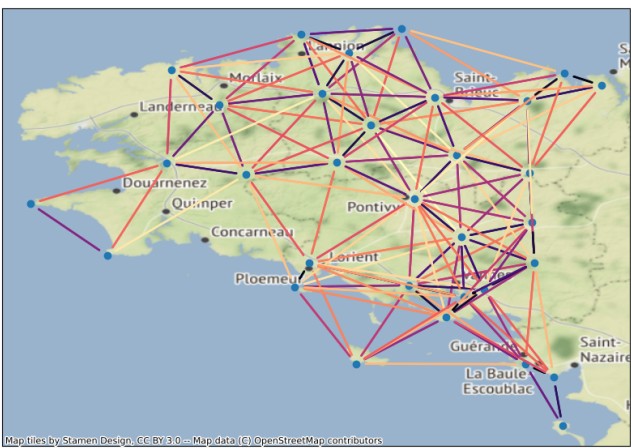

Figure 2: Prior graph for the multisensor network example.

**Intracardiac electrocardiograms:** In this scenario, we are measuring electrical activities in the heart using multiple electrodes. While this example falls within the case of multi-sensor networks, and the prior graph can be constructed as previously explained, $\mathbf{A}^{\mathrm{prior}}$ can also be computed by focusing on the similarity between signals recorded by the electrodes, using cross-correlation for example. It allows guiding the algorithm to find causality links between the most similar signals (cf Figure 3).

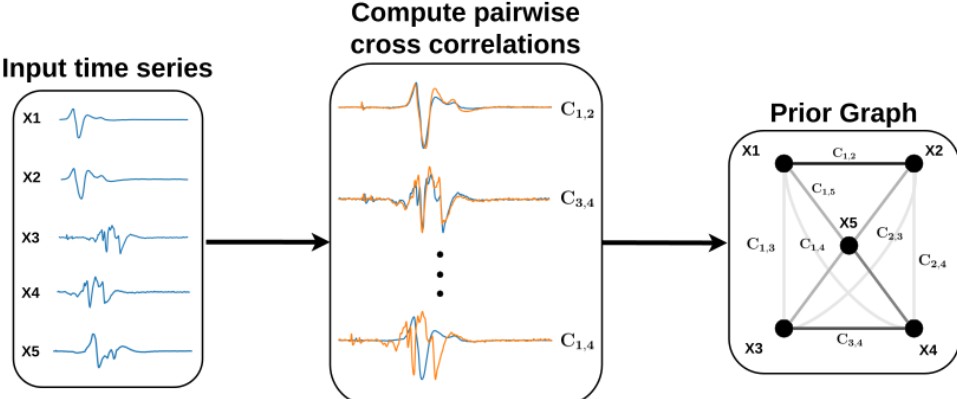

Figure 3: Construction of the graph prior for intracardiac signals example.

### 3.3 Links with existing models

The work of Yao et al. (2013) is the closest one to ours. They propose to leverage prior knowledge $\mathbf{A}^{\mathrm{prior}} \in \mathbb{R}^{p \times p}$ solving:

$$\hat{\mathbf{C}} = \underset{\mathbf{C}}{\arg\min} \frac{1}{2} \|X[t] - \mathbf{C}X[t-1]\|_F^2 + \frac{1}{2}\lambda_1 N(\mathbf{C} - \lambda_2 \mathbf{A}^{\mathrm{prior}}) \ , \tag{8}$$

with $N : M \mapsto \|M\|_F$ or $N : M \mapsto \|M\|_1$. However this model works under the assumption that we have prior knowledge about the exact values of the matrix $\mathbf{C}$, which can be unrealistic in some applications. Consequently, this framework impose to know the sign of values in $\mathbf{C}$ since the optimization problem penalizes the component wise difference between $\mathbf{C}$ and $\mathbf{A}^{\mathrm{prior}}$. The authors propose to address this issue by hand, computing a Ridge estimator of the VAR model to chose the signs of $\mathbf{A}^{\mathrm{prior}}$. On the contrary, by exploiting a weighted Lasso, our model does not suffer from these limitations and only require relative prior knowledge about relationships.

Our method is also closed to an Adaptive Lasso estimator obtained with:

$$\min_{\mathbf{C}} \quad \frac{1}{N} \sum_{n=1}^{N} \left\| X^{(n)}[t] - \mathbf{C} X^{(n)}[t-1] \right\|_2^2 + \lambda \sum_{1 \le i < j \le p} \frac{|\mathbf{C}_{i,j}| + |\mathbf{C}_{j,i}|}{\mathbf{A}_{i,j}^{\mathrm{prior}}} \ . \tag{9}$$

Indeed, considering only the two first terms of our problem, we obtain exactly the Adaptive Lasso one. However the Adaptive Lasso estimator is not robust to the errors in the weights, since the asymptotic consistency is achieved taking for weights unbiased estimators of the true solution (Zou, 2006). Thus, the last term in the optimization problem allows to refine the prior through iterations using information in the time series resulting in a more robust version of Adaptive Lasso as will be seen in the experiments (Section 4).

### 3.4 Solving method: A-AdaptiveLasso (AALasso)

The function to minimize in (7) is not convex in $(\mathbf{A}, \mathbf{C})$. Indeed, we can show that even with $\mathbf{C}$ fixed, the function in $\mathbf{A}$ is not convex (c.f. Appendix A). However, the function is convex in $\mathbf{C}$ with $\mathbf{A}$ fixed (Adaptive Lasso problem) and we have a closed form for the roots of the derivative in $\mathbf{A}$ (with $\mathbf{C}$ fixed). Thus, we use an alternating minimization algorithm to solve this problem.

#### 3.4.1 C update.

For fixed $\mathbf{A}$, the optimization problem (7) with respect to $\mathbf{C}$ is:

$$\min_{\mathbf{C}} \quad \frac{1}{N} \sum_{n=1}^{N} \left\| X^{(n)}[t] - \mathbf{C} X^{(n)}[t-1] \right\|_2^2 + \lambda \sum_{1 \le i < j \le p} \frac{|\mathbf{C}_{i,j}| + |\mathbf{C}_{j,i}|}{\mathbf{A}_{i,j}} \ . \tag{10}$$

From Eq. (10), we see that the optimization step in $\mathbf{C}$ is an Adaptive Lasso problem with weights equal to $1/\mathbf{A}_{i,j}$ (Zou, 2006). A common way to solve Adaptive Lasso regression is to remark that it can be written like a Lasso problem. Indeed, considering $\tilde{\mathbf{C}}_{i,j} = \frac{\mathbf{C}_{i,j}}{A_{i,j}}$, we can write the problem as a Lasso one transforming $\mathbf{X}$.

While there is no closed formula to compute the Lasso estimator in the general case, common ways to solve it are to use: (i) the least-angle regression (LARS) algorithm Efron et al. (2004), (ii) algorithms based on coordinate descent Wu & Lange (2008), or (iii) the ADMM algorithm Boyd (2010). A recent survey presents some of these algorithms and provides their convergence rates (Zhao & Huo, 2023). Details of the ADMM updates to solve Problem 10 are presented below. Let $i \in [\![1, p]\!]$, we first rewrite (10) as $p$ subproblems:

$$\min_{\mathbf{C}_{i:}} \quad \frac{1}{N} \left\| X_i[t] - \mathbf{C}_i X[t-1] \right\|_2^2 + \lambda \sum_{j=1}^{p} \frac{|\mathbf{C}_{i,j}|}{\mathbf{A}_{i,j}} \ , \quad i = 1, ..., p \ , \tag{11}$$

with $X_i[t]$ and $X_i[t-1]$ the vectors containing the $N$ samples of the variable $i$. Then we rewrite (11) as a Lasso problem:

$$\min_{\tilde{\mathbf{C}}_{i:}} \quad \frac{1}{N} \left\| X_i[t] - \tilde{\mathbf{C}}_i \tilde{X}[t-1] \right\|_2^2 + \lambda \sum_{j=1}^{p} |\tilde{\mathbf{C}}_{i,j}| \ , \quad i = 1, ..., p \tag{12}$$

where $\tilde{\mathbf{C}}_{i,j} = \frac{\mathbf{C}_{i,j}}{\mathbf{A}_{i,j}}$ and $\tilde{X}[t-1] = (A_{i,j} \times X_{i,j})_{1 \le j \le p}$. In order to use the ADMM algorithm, we rewrite the problem as follows:

$$\min_{\tilde{\mathbf{C}}_{i:}, U} \frac{1}{N} \left\| X_i[t] - \tilde{\mathbf{C}}_i \tilde{X}[t-1] \right\|_2^2 + \lambda \|U\|_1 \quad \text{subject to} \quad \tilde{\mathbf{C}}_{i:} - U = 0 \ .$$

Finally, the ADMM updates are:

$$\tilde{\mathbf{C}}_i^+ = \arg\min_{\tilde{\mathbf{C}}_i} \frac{1}{2} \|X_i[t] - \tilde{\mathbf{C}}_i \tilde{X}[t-1]\|_2^2 + \frac{\rho}{2} \|\tilde{\mathbf{C}}_i - U + W\|_2^2$$

$$= (\tilde{X}[t-1]\tilde{X}[t-1]^T + \rho I)^{-1}(X_i[t]\tilde{X}[t-1]^T + \rho(U - W))$$

$$U^+ = \arg\min_U \lambda\|U\|_1 + \frac{\rho}{2}\|\tilde{\mathbf{C}}_i^+ - U + W\|_2^2$$

$$= S_{\lambda/\rho}(\tilde{\mathbf{C}}_i^+ + W) \quad \text{(Soft-thresholding of } \tilde{\mathbf{C}}_i^+ + W)$$

$$W^+ = W + \tilde{\mathbf{C}}_i^+ - U^+$$

$$\text{where} \quad [S_t(x)]_j = \begin{cases} x_j - t & \text{if } x > t \\ 0 & \text{if } -t \leq x \leq t, \quad j = 1,\ldots,p \\ x_j + t & \text{if } x < -t \end{cases}$$

and $W, U$ are auxiliary variables specific to the ADMM algorithm.

### 3.4.2 A update.

For fixed $\mathbf{C}$, the optimization problem (7) with respect to $\mathbf{A}$ is:

$$\min_{\mathbf{A}} \quad \lambda \sum_{1 \leq i < j \leq p} \frac{|\mathbf{C}_{i,j}| + |\mathbf{C}_{j,i}|}{\mathbf{A}_{i,j}} + 2\lambda \sum_{1 \leq i < j \leq p} \log(2\mathbf{A}_{i,j}) + \gamma \|\mathbf{A} - \mathbf{A}^{\text{prior}}\|_F^2 , \tag{13}$$

$$\text{subject to} \quad \mathbf{A}_{i,j} \geq 0 , \quad \mathbf{A}_{i,j} = \mathbf{A}_{j,i} , \quad 1 \leq i < j \leq p .$$

To address the symmetry constraint, a straightforward way is to optimize over the upper diagonal values and to set $\mathbf{A}_{j,i} = \mathbf{A}_{i,j}$ for $i < j$. The minimisation can then be carried out by directly calculating the exact minimum, which is given in the next proposition.

**Proposition 7.** *The roots of the derivative with respect to $\mathbf{A}_{l,m}$ of the objective function* (13) *are:*

$$z_k = \frac{\mathbf{A}_{l,m}^{prior}}{3} + e^{2ik\pi/3} \sqrt[3]{\frac{1}{2}\left(-q + \sqrt{\frac{\Delta}{27}}\right)} + e^{-2ik\pi/3} \sqrt[3]{\frac{1}{2}\left(-q - \sqrt{\frac{\Delta}{27}}\right)} , \qquad k = 0, 1, 2 , \tag{14}$$

$$\text{where} \quad \begin{cases} p = -\frac{(\mathbf{A}_{l,m}^{prior})^2}{3} + \frac{\lambda}{2\gamma} \\ q = -\frac{\mathbf{A}_{l,m}^{prior}}{3}\left(\frac{8\gamma(\mathbf{A}_{l,m}^{prior})^2}{9} - 2\lambda\right) - \lambda\left(|\mathbf{C}_{l,m}| + |\mathbf{C}_{m,l}|\right) \\ \Delta = 4p^3 + 27q^2 . \end{cases}$$

*Furthermore, there exists at least one positive root for the derivative with respect to $\mathbf{A}_{l,m}$, and the global minimum on the interval $]0, +\infty[$ is attained at one of these roots.*

*Proof of 7.* Finding the roots of the objective function derivative with respect to $\mathbf{A}_{l,m}$ is equivalent to finding the roots of a 3-degree polynomial, allowing us to apply the Cardan formula. Subsequently, as the objective function in $\mathbf{A}_{l,m}$ diverges towards infinity at the boundaries of $]0, +\infty[$, there exists at least one positive root, and the minimum is attained at one of these roots. $\square$

### 3.4.3 Alternating Minimization algorithm.

The final AALasso algorithm is presented in Algorithm 1. AALasso depends on four input parameters:

(i) $\lambda$ controls the sparsity of the learned graph. The larger $\lambda$, the more sparse the solution.

(ii) $\gamma$ controls the confidence in the prior graph. A large $\gamma$ will constrain $\mathbf{A}$ to stay close to $\mathbf{A}^{\text{prior}}$, whereas a small value allows $\mathbf{A}$ to deviate from $\mathbf{A}^{\text{prior}}$.

(iii) The choice of $\mathbf{A}^{(0)}$ has an impact on the solution since we start by solving the problem in $\mathbf{C}$ with fixed $\mathbf{A}$. The straightforward idea is to take $\mathbf{A}^{(0)} = \mathbf{A}^{\text{prior}}$, but note that other initializations can be chosen (c.f. Appendix B.3). For instance, AALasso can be seen as a generalization of Lasso and

---

**Algorithm 1:** Fitting algorithm.

---

**input** : $N_{\text{iter}}$, $\lambda$, $\gamma$, $\mathbf{A}^{\text{prior}}$
**output:** $\widehat{\mathbf{C}}$, $\widehat{\mathbf{A}}$
$\mathbf{A}^{(0)} \leftarrow \mathbf{A}^{\text{prior}}$
**for** $r \leftarrow 1$ **to** $N_{iter}$ **do**
$\quad \mathbf{C}^{(r)} \leftarrow f_{\mathbf{C}}(\mathbf{C}, \mathbf{A}^{(r-1)})$ where $f_{\mathbf{C}}$ denotes the update in (3.4.1).
$\quad \mathbf{A}^{(r)} \leftarrow f_{\mathbf{A}}(\mathbf{C}^{(r)}, \mathbf{A})$ where $f_{\mathbf{A}}$ denotes the update in (3.4.2).
**return** $\mathbf{C}^{(N_{\text{iter}})}, \mathbf{A}^{(N_{\text{iter}})}$.

---

LS+ALasso since these two estimators correspond to the first iteration of AALasso for particular choices of $\mathbf{A}^{(0)}$. Indeed, taking $\mathbf{A}^{(0)}$ equals to the one matrix $(\mathbf{1})_{1 \leq i,j \leq p}$, the $\mathbf{C}$ update corresponds to the Lasso algorithm, while taking $\mathbf{A}^{(0)} = (w_{i,j})_{1 \leq i,j \leq p}$ where $w_{i,j}$ are the weights computed by the Least Square Estimator, the first step corresponds to the LS+ALasso estimator.

(iv) The number of iterations $N_{\text{iter}}$ of the alternating minimization algorithm impact directly the runtime and the performances, and in practice we will chose a relatively small number of iterations (around 10), according to synthetic experiments conducted in 4.

**Remark 8.** *Note that if it exists an iteration $r_0$ such that $\mathbf{A}_{i,j}^{(r_0)} = 0$, the coefficients $\mathbf{C}_{i,j}^{(r)}$ and $\mathbf{C}_{j,i}^{(r)}$ will be zero for $r > r_0$. Next, in order to allow the algorithm to add an edge that is not originally present in the prior graph, we set the minimum values of the adjacency matrix to $\epsilon > 0$.*

**Remark 9.** *As explained in Remark 6, it is possible to leverage a directed prior graph with slights modifications of the model. Regarding the solving method, the $\mathbf{C}$ update remains the same, and for the $\mathbf{A}$ update, we compute the solutions of $p^2$ 3-degree polynomial rather than $p(p-1)/2$.*

### 3.5 Theoretical properties

In this section, we prove the convergence towards a set of stationary points of our algorithm 1 to solve the optimization problem (7), and we show that the time complexity is asymptotically in the same order than the one of Lasso estimator.

#### 3.5.1 Convergence

Classical results of alternating minimization convergence assume that the objective function is differentiable (c.f. Grippo & Sciandrone (2000)). However it is not applicable to our case since our objective function in (7) is not differentiable in $\mathbf{C}_{i,j} = 0$. We now introduce the lower directional derivatives to address this issue.

**Definition 10** (Lower directional derivative)**.** *For any $x \in \mathbb{R}^p$ and any $v \in \mathbb{R}^p$, we denote the (lower) directional derivative of $f$ at $x$ in the direction $v$ by $f'(x;v) = \lim\inf_{\lambda \downarrow 0} \left[ \frac{f(x + \lambda v) - f(x)}{\lambda} \right]$.*

**Definition 11** (Stationary points)**.** *We say that $z$ is a **stationary point** of $f$ if $z \in dom f$ and $f'(z;v) \geq 0$, $\forall v$.*
*We say that $z$ is a **coordinate-wise minimum** point of $f$ if $z \in dom f$ and $f(z + (0,...,v_k,...,0)) \geq f(z)$, $\forall v_k \in \mathbb{R}^{n_k}$, for all $k = 1,...,N$.*

Using this framework, the following theorem holds.

**Theorem 12.** *(Convergence of AALasso)*
*The sequence $\left\{ (\mathbf{C}^{(r)}, \mathbf{A}^{(r)}) \right\}_{r=1,2,...}$ generated by the two-blocks alternating minimization is well defined and bounded. Moreover, every cluster point is a stationary point of the problem 7.*

*Proof.* The proof is given in Appendix A.2. $\qquad\square$

### 3.5.2 Computational complexity

Recall that when fitting a Lasso estimator for learning VAR parameters in $p$ dimension, we fit $p$ independent Lasso estimators in dimension $p$. In the following, $p$ denotes the dimension and $N$ the number of samples. Recall that our algorithm iteratively solves two steps, $\mathbf{A}$ and $\mathbf{C}$. First, as detailed in 3.4.1, the $\mathbf{C}$ update is solved by transforming the Adaptive Lasso problem into a Lasso problem in $O(p \times N)$ and using ADMM to fit the Lasso parameters in $O(p^3 + N \times p^2)$ (see A.3 for more details). Thus, the time complexity of the $\mathbf{C}$ update is in $O(p^3 + N \times p^2)$. Then the $\mathbf{A}$ update is done computing a closed formula in $O(1)$ for each value $\mathbf{A}_{i,j}$, $1 \leq i < j \leq p$, so this step is in $O(p^2)$, which is negligible compared to the $\mathbf{C}$ update. Finally, considering a fixed number of iterations $N_{\text{iter}}$, the AALasso algorithm has a time complexity in $O(N_{\text{iter}} \times p^3 + N_{\text{iter}} \times N \times p^2)$, which is approximately $N_{\text{iter}}$ times the one of the Lasso estimator solved with the ADMM algorithm.

### 3.6 Link with Statistical Model

It is interesting to adopt the Bayesian point of view to deeper understand the statistical hypothesis behind the model and how the algorithm works. Here, it can be shown that the optimization problem (7) presented in the previous section is equivalent to the Maximum A Posteriori (MAP) of the probabilistic graphical model presented above in Eq. (15).

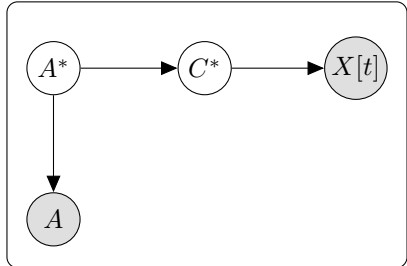

with :

$$\begin{cases} \mathbf{A}_{i,j} \sim \mathcal{N}(\mathbf{A}_{i,j}^*, \sigma^2) \ , \ i, j = 1, ..., p \\ \mathbf{C}_{i,j}^* \sim \text{Laplace}(0, \mathbf{A}_{i,j}^*) \ , \ i, j = 1, ..., p \\ X[t] \sim \mathcal{N}(\mathbf{C}^* X[t-1], \sigma_X^2) \end{cases} \tag{15}$$

Figure 4: Observable variables are in grey and latent variables in white.

**Theorem 13.** *The solutions of Problem* (7) *correspond to the Maximum A Posteriori (MAP) of the statistical model* (15) *under the assumption that* $\mathbf{A}^*$ *follows the improper distribution* $\mathbf{1}_{S_p(\mathbb{R})}$.

*Proof of 13.* The proof is given in Appendix A.1. □

Then, the normalization term $\log(2\mathbf{A}_{i,j})$ introduced in (7) corresponds to the normalization of the Laplace distribution. Intuitively, it allows $\mathbf{A}$ to remain meaningful regarding the Laplace distribution and it avoids parameters to go to infinity.

This probabilistic graphical model allows a good understanding of the model introduced. An unknown graph $\mathcal{G}^*$ first generates a parameter matrix $\{\mathbf{C}_{i,j}^*\}_{i,j}$ from Laplace distribution with variances equal to $\{\mathbf{A}_{i,j}^*\}_{i,j}$ (note that Laplace distribution encourages sparsity), and $\mathbf{C}^*$ allows to generate the multivariate time series $X$ following a VAR(1) model. On the other side, we observe a noisy (Gaussian noise) version of $\mathbf{A}^*$. To

leverage the information provided by $\mathbf{A}$ to learn $\mathbf{C}^*$ we propose to jointly infer the two latent variables $\mathbf{A}^*$ and $\mathbf{C}^*$.

### 3.7 Generalization to VAR($d$) models

In the preceding section, we discussed VAR(1) models for simplicity. However, certain applications require the consideration of VAR($d$) models and we detail the generalization of our model in this section. A VAR($d$) model can be expressed as a VAR(1) model as follows. Let $X$ a process defined by a VAR($d$) model:

$$X[t] = \sum_{\tau=1}^{d} \mathbf{C}^\tau X[t - \tau] + \varepsilon[t] ,$$

where $X[t] = (X_1[t], ..., X_p[t])$ is a random $p$-dimensional time series and $\varepsilon[t] \sim \mathcal{N}(0, \sigma_X^2 I_p)$, $\sigma_X > 0$. For

$t \geq d$, let $\overline{\mathbf{X}}[t] = (X[t], X[t - 1], ..., X[t - d + 1])^T$ and $\overline{\mathbf{C}} = \begin{bmatrix} \mathbf{C}^1 & \mathbf{C}^2 & \cdots & \mathbf{C}^d \\ 1 & 0 & \cdots & 0 \\ \vdots & \ddots & \ddots & \vdots \\ 0 & \cdots & 1 & 0 \end{bmatrix}$, then the process $\overline{\mathbf{X}}[t]$

satisfies the VAR(1) model:

$$\overline{\mathbf{X}}[t] = \overline{\mathbf{C}}\,\overline{\mathbf{X}}[t-1] + (\epsilon[t], ..., \epsilon[t - d + 1])^T .$$

Note that the stability assumption of the VAR model is now satisfied if $\rho(\overline{\mathbf{C}}) < 1$ (cf (Lütkepohl, 2005)). Using this last point, our model is still applicable for some $d \in \mathbb{N}$ assuming we have a prior matrix defined by:

$$\overline{\mathbf{A}^{\mathrm{prior}}} = \begin{bmatrix} \mathbf{A}_1^{\mathrm{prior}} & \mathbf{A}_2^{\mathrm{prior}} & \cdots & \mathbf{A}_d^{\mathrm{prior}} \\ 1 & 0 & \cdots & 0 \\ \vdots & \ddots & \ddots & \vdots \\ 0 & \cdots & 1 & 0 \end{bmatrix} .$$

Thus, solving this problem with a number of lags $d > 1$ is equivalent to solve a problem with 1 lag in dimension $p \times d$, since we can consider the lags $> 1$ as new variables.
Note that a straightforward idea is to set $\mathbf{A}_l^{\mathrm{prior}} = \mathbf{A}^{\mathrm{prior}}$ for $l = 1, ..., d$ but this generalization allows to leverage prior knowledge about relationships likelihood at each order.

## 4  Experiments

In this section, a large series of experiments is carried out to assess the effectiveness and applicability of AALasso to learn Granger causality graphs. Our algorithm is tested using both synthetic and real-world data sets. To utilize AALasso effectively, the input requirements include a multivariate time series and the availability of an adjacency matrix representing a prior network structure. By employing synthetic and real datasets, we evaluate AALasso's robustness across various scenarios, including limited number of samples and several levels of noise, and real-world datasets. These experiments provide valuable insights into the algorithm's capabilities and its potential applications in diverse domains. The code to reproduce our experiments with synthetic data will be available.

### 4.1  Task and evaluation metrics

In these experiments, the objective is to learn a Network Granger causality (NGC) from given multivariate time series and a prior network. We suppose that these series follow a VAR(1) model, hence, learning the NGC is equivalent to fit the VAR parameters, i.e., given $X \in \mathbb{R}^{p \times N}$ and $\mathbf{A}^{\mathrm{prior}}$ in $\mathbb{R}^{p \times p}$ we want to estimate the matrix $\mathbf{C}$.
Since VAR models are usually employed for forecasting tasks, a standard metric to evaluate estimators is the normalized Root Mean Square Error (nRMSE) of the one step predictions. Let $X[t]$ be a multivariate

time series and $\hat{X}[t]$ be the reconstruction using the fitted VAR model at time $t$, the nRMSE is defined by:

$$\text{nRMSE}(\hat{X}) := \sqrt{\frac{\sum_t \left\| \hat{X}[t] - X[t] \right\|_2^2}{\sum_t \|X[t]\|_2^2}}.$$

Note that, the main objective of this paper is to learn the underlying NGC, so we are more interested in learning a relevant graph than allowing a good reconstruction (even though the two tasks are correlated). To evaluate the quality of the learned graph, we compute the F1-score between $\hat{\mathbf{C}}$ and $\mathbf{C}^*$ (see e.g. (Pasdeloup et al., 2016) for more information). This metric is defined as follows:

$$\text{F1-score} := \frac{2 \cdot \text{precision} \cdot \text{recall}}{\text{precision} + \text{recall}},$$

where

$$\text{precision} = \frac{\text{True Positives}}{\text{True Positives} + \text{False Positives}}$$

and

$$\text{recall} = \frac{\text{True Positives}}{\text{True Positives} + \text{False Negatives}}.$$

Here, the precision measures the proportion of correctly identified causality links, while recall measures the ability to capture all causality links. Note that the F1-score is only available for synthetic data as we need to have access to the true graph (the true VAR model).

Although these two measures are related (a good graph should lead to a good reconstruction), it should be noted that a good reconstruction can be achieved by a relatively dense graph. Given that sparsity is a desired property, the F1 score is used to understand whether the learned graph can efficiently reconstruct time series while avoiding irrelevant edges.

## 4.2 Methods

The aim of these experiments is to show that AALasso can exploit prior knowledge to improve its performance compared to existing methods. We compare our estimator to the classical estimators: the Lasso and the Adaptive Lasso with weights equal to the least squares estimates (noted LS + ALasso, cf (Zou, 2006)). Moreover, since the first step of our algorithm is equivalent to solve an Adaptive Lasso problem assuming that the weights are given by $W_{i,j} = \frac{1}{\mathbf{A}_{i,j}^{\text{prior}}}$, we compare our method with this first step (denoted 1-AALasso) to demonstrate the usefulness to perform several steps. Note that we do not show the performances of the Least Squares estimator since the results are poor in settings with only few samples. Finally, note that the Lasso and LS+ALasso algorithms do not take into account the prior matrix, so the prior noise will not impact their results.

**Implementation details.** For all experiments, we used the package `asgl` (Álvaro Méndez Civieta et al., 2021) implemented using `cvxpy` (Diamond & Boyd, 2016) to solve Lasso and Adaptive Lasso regression problems.

## 4.3 Datasets

### 4.3.1 Synthetic Data

Synthetic data are generated with respect to the statistical model (15). To define the matrix $\mathbf{A}^*$, we first generate $p = 40$ points in $[0,1]^2$ uniformly at random. Then, we construct a matrix $\mathbf{D} \in \mathbb{R}_*^{p \times p}$ by applying the Gaussian kernel to the pairwise Euclidean distance between the points (the standard deviation of the Gaussian kernel is taken equal to the median values of all pairwise distances). $\mathbf{A}^*$ is obtained by randomly setting to 0 a ratio $\tau_m = 0.5$ of values of $\mathbf{D}$ (mispecified edges) and cutting to 0 values smaller than $\tau = 0.7$ to promote sparsity. Finally, the VAR parameters are drawn from $Laplace(0, \mathbf{A}_{i,j}^*)$ and $\mathbf{A}^{\text{prior}} = \mathbf{D} + \mathcal{E}$, where $\mathcal{E}$ is a symmetric matrix whose subdiagonal values are sampled from i.i.d. Gaussian distribution with

---

**Algorithm 2:** Data generation.

---

**input** : $p, \tau_m, \tau, \sigma_X, \sigma_A$
**output:** $\mathbf{A}^{\text{prior}}, X$
Generate randomly $p$ points in $[0,1]^2$.
Compute the euclidean distance matrix $\mathbf{D}$.
$\mathbf{A}^*$ is obtained setting to 0 a ratio of $\tau_m$ values of $\mathbf{D}$ (mispecified edges).
Generate VAR parameters $\mathbf{C}^*_{i,j} \sim Laplace(0, \mathbf{A}^*_{i,j})$.
**for** $1 \leq i < j \leq p$ **do**
  $\lfloor$ Randomly set $\mathbf{C}^*_{i,j}$ or $\mathbf{C}^*_{j,i}$ to 0 (directed graph).
$\mathbf{A}^{\text{prior}} = \mathbf{D} + \mathcal{E}$ where $\mathcal{E}$ is a symmetric matrix where subdiagonal values are sampled from i.i.d.
 gaussian distribution with variance $\sigma_A^2$.
Sample $X[t] \sim \mathcal{N}(\mathbf{C}^* X[t-1], \sigma_X^2)$
**return** $\mathbf{A}^{\text{prior}}, X$

---

variance $\sigma_A^2$ (varying in $\{0.02, 0.1, 0.25, 0.35\}$ to test several level of noises). Note that $\mathbf{A}^{\text{prior}}$ is computed from $\mathbf{D}$ and not from $\mathbf{A}^*$. Indeed, $\mathbf{A}^*$ is obtained by performing sparse variable selection from $\mathbf{D}$; therefore, we do not incorporate this variable selection into $\mathbf{A}^{\text{prior}}$ and the objective is to assess the effectiveness of AALasso in accurately retrieving this selection. This data generation is summarized in Algorithm (2). At the end, for each experiment, we sample $N = \{80, 200, \ldots, 500\}$ different time series $X[t] \sim \mathcal{N}(\mathbf{C}^* X[t-1], \sigma_X^2)$, $\sigma_X^2 = 0.1$, which we split into training and test sets of equal sizes. We repeat this procedure 20 times for each value of $N$.

### 4.3.2 Breast Cancer Network

**Dataset** The Heritage Provider Network DREAM 8 Breast Cancer Network Prediction dataset focuses on predicting causal protein networks using time series data from reverse phase protein array (RPPA) experiments. The aim is to advance breast cancer understanding by using complex time series data and computational modeling to uncover causal relationships within protein networks responding to various stimuli and inhibitors across different cell lines. It involves examining four cell lines (BT549, BT20, MCF7, and UACC812) under four inhibitor conditions (AKT, AKT + MET, FGFR1 + FGFR3, and DMSO control) and eight ligand stimuli (Serum, PSB, EGF, Insulin, FGF1, HGF, NRG1, and IGF1) at multiple time points (t = 0, 5 min, 15 min, 30 min, 1 hr, 2 hr, and 4 hr). Here, the task is to create 32 causality networks, one for each combination of stimulus ligand and cell line, from protein probe across the time points. Formally, given a 3-uple (cell line, ligand, inhibitor), we observe a multivariate times series $X \in \mathbb{R}^{p \times N}$ with $N = 6$ and $p \in \{41, 45, 48\}$ and we want to learn a graph of Granger causalities $\mathcal{G}$.

**Choice of $\mathbf{A}^{\text{prior}}$** In (Carlin et al., 2017), the authors successfully utilized a prior network derived from the Pathway Commons database version 3 Cerami et al. (2010) to enhance their analysis. In essence, this network prior was developed by aggregating various established pathways from Pathway Commons, where protein interactions closely aligned with the concept of causal influence. It assumed that interactions declared in these pathway databases implied that perturbations to upstream regulatory proteins could lead to either direct or indirect perturbations of downstream target proteins connected via directed paths. The adjacency matrix was obtained performing a heat diffusion over the initial graph (cf (Carlin et al., 2017) for more details). Importantly, the network prior was independent of training data and could be reused across experiments. The utilization of this network prior improved performances, highlighting the importance of taking into account prior knowledge for the causality inference task. However, in (Carlin et al., 2017) the authors did not take into account this network in their inference algorithm: they averaged a posteriori the output of the GENIE3 (Huynh-Thu et al., 2010) algorithm with the adjacency matrix of this prior. We therefore chose $\mathbf{A}^{\text{prior}}$ equals to this adjacency matrix.

Note that the objective of this challenge is to infer causality but not necessarily temporal/Granger causality. Thus, we do not expect to achieve better results than the challengers but the objective is to demonstrate that taking into account a prior can be relevant.

**Pre-processing of time series** We normalize the time series as a pre-processing step to satisfy the first-order stationary assumption:

$$X[1:T] \leftarrow \frac{X[1:T] - \overline{X}}{\sigma_X} \tag{16}$$

where $\overline{X}$ is the mean of $X$ and $\sigma_X$ its standard deviation.

### 4.3.3 Molène Dataset

**Dataset** The Molène datase contains hourly temperatures recorded by sensors at $p = 32$ locations in Britany (France) during $N = 746$ hours. Here the objective is to understand the spatio-temporal dynamics of the temperature and to assess the extent to which the model can describe complex phenomena (such as weather) by considering only data and geographical information (sensor positions).

**Choice of $\mathbf{A^{prior}}$** The prior adjacency matrix is $\mathbf{A}_{i,j}^{\mathrm{prior}} = \exp(-dist(i,j)/\overline{dist})$ where $dist(i,j)$ is the Euclidean distance between the stations $i$ and $k$ and $\overline{dist}$ is the median of all computed distances. Note that this case is a good example of what could be a "noisy" prior knowledge, since the euclidean geometry is isotropic contrarily to the weather dynamics.

**Pre-processing of time series** We consider the first derivative of the signals rather than original signals in order to verify as much as possible the wide-sense stationary property.

### 4.4 Results on synthetic data

### 4.4.1 Comparison with classical algorithms

In this part, we compare AALasso with standard method to compute Network Granger Causality. Thus we focus on methods based solving the classic NGC optimization problem with some regularization (Lasso, Adaptive Lasso). The aim of these experiments is to show that our method is able to leverage prior knowledge to increase the accuracy of the retrieved NGC.

For all of the 20 experiments, we performed $N_{\mathrm{iter}} = 10$ (see 4.4.4) iterations in the alternating minimization algorithm (1) using half of the training set. The parameters $\lambda$ and $\gamma$ were selected by cross-validation minimizing the nRMSE over the second half of the training set (the validation set).

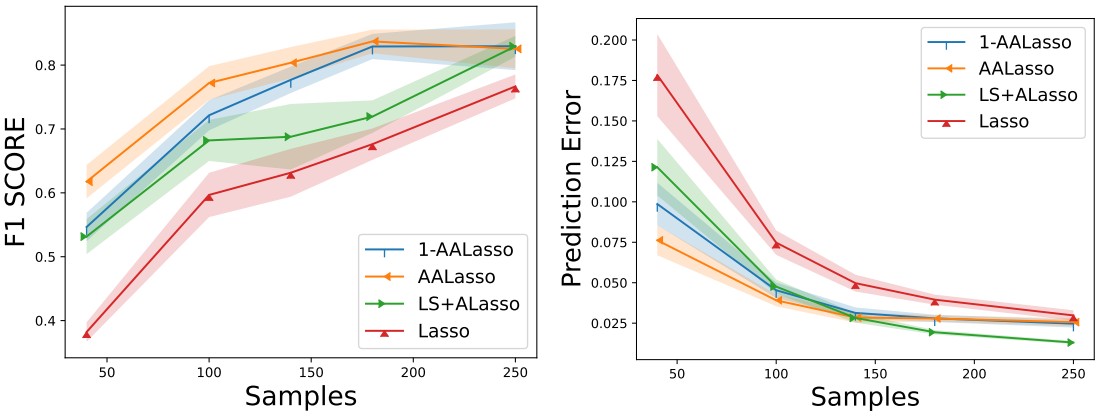

Figure 5: rNMSE and F1-score in function of the number of samples used for training using $\sigma_A = 0.1$. We plotted the 90% confidence intervals.

The results in Figure 5 exhibit better reconstruction and greater F1-score when utilizing AALasso rather than vanilla methods when the number of samples is lower than 140. From 40 to 140 samples, AALasso

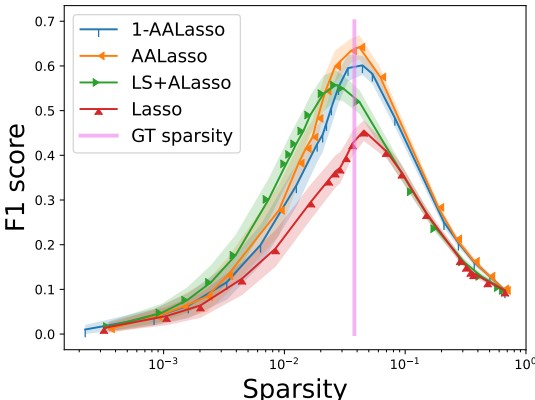

Figure 6: F1-score in function of the sparsity of the learned graph $s = \frac{\text{number of edges}}{p(p-1)}$ (obtained using several values for $\lambda$) with $\sigma_A = 0.1$ and $N = 40$. A sparsity of 0 means an empty graph while an sparsity of 1 means a dense graph. We plotted the 90% confidence intervals.

returns F1-scores from 0.6 to 0.8 while LS+ALasso F1-score ranges from 0.5 and 0.73 (an average gain of 0.1). Thus, our algorithm effectively leverages the additional information in settings with few samples, and our approach enables fine-tuning of the graph while remaining a good forecasting power. Results presented in Table 2 provides more details regarding the computation of F1-score (precision and recall). We observe that the gain in the F1-score is a consequence of a important gain in precision. Indeed, whereas the first iteration of the algorithm results in relatively good recall and precision, the next iterations lead to important improvements of precision with limited loss in recall. Moreover, the results shown in Figure 6 indicate that AALasso outperforms Lasso and LS+ALasso estimators for sparsity levels near to the ground truth one (with a manual selection of $\lambda$). These results confirm that the model allows to increase the performances of the variable selection and that the gains in Figure 2 are not just a consequence of a better selection of $\lambda$.

When the number of samples increases, the LS+ALasso estimator provides better reconstruction than AALasso. This behavior can be explained by the fact that the prior knowledge given is noisy so the AALasso estimator is biased. In practice it allows to reduce the variance when few samples are available, but when the number of samples becomes large enough to perform statistical inference directly from time series data, this bias lead to slightly less accurate results. Thus, the question is to know whether the time series are informative to know if adding biased prior knowledge will improve or not the results. However, recall that we are interested in a graph learning task so the F1-score is more informative than the reconstruction error, and shows satisfying results even for relatively large number of samples (until a certain threshold, here 250 samples). This can be explained by the fact that the algorithm prefers precision to recall, and will focus on returning true causal relations, sacrifying reconstruction and recall. Finally, the difference of results between the first iteration and the complete optimization process of AALasso points out the interest of the alternating minimization.

| N Samples | 40 | | | 100 | | | 180 | | |
|---|---|---|---|---|---|---|---|---|---|
| Metrics | P | R | F1 | P | R | F1 | P | R | F1 |
| **Lasso** | 0.28 (0.04) | 0.61 (0.09) | 0.38 (0.04) | 0.51 (0.1) | 0.75 (0.08) | 0.6 (0.09) | 0.57 (0.08) | 0.83 (0.06) | 0.68 (0.07) |
| **LS+ALasso** | 0.54 (0.09) | 0.54 (0.11) | 0.53 (0.07) | 0.69 (0.1) | 0.68 (0.1) | 0.68 (0.09) | 0.64 (0.08) | 0.83 (0.06) | 0.72 (0.07) |
| **1-AALasso** | 0.44 (0.06) | **0.73** (0.1) | 0.55 (0.06) | 0.65 (0.06) | **0.82** (0.09) | 0.72 (0.06) | 0.83 (0.06) | **0.84** (0.06) | 0.83 (0.05) |
| **AALasso** | **0.55** (0.07) | 0.71 (0.11) | **0.62** (0.07) | **0.76** (0.06) | 0.79 (0.1) | **0.77** (0.07) | **0.87** (0.05) | 0.81 (0.07) | **0.84** (0.05) |

Table 2: Precision, Recall and F1-score in function of the number of samples. We took a noise over the prior network with $\sigma_A = 0.1$.

**Experiments in higher dimension** Finally, to understand whether our method was able to deal with high-dimensional environment (i.e $N << p$), we conducted the same experiments taking $p \in \{60, 100, 160\}$ and $N = 40$ (still 20 samples for training and 20 for validation).

| Dimension | 60 | | | 100 | | | 160 | | |
|-----------|------|------|------|------|------|------|------|------|------|
| Metrics | P | R | F1 | P | R | F1 | P | R | F1 |
| **Lasso** | 0.31 (0.07) | 0.48 (0.09) | 0.37 (0.07) | 0.26 (0.04) | 0.32 (0.06) | 0.28 (0.04) | 0.26 (0.03) | 0.24 (0.04) | 0.25 (0.03) |
| **LS+ALasso** | **0.58** (0.06) | 0.38 (0.10) | 0.45 (0.08) | **0.52** (0.06) | 0.20 (0.05) | 0.29 (0.05) | **0.48** (0.05) | 0.13 (0.04) | 0.20 (0.05) |
| **1-AALasso** | 0.45 (0.05) | **0.60** (0.10) | 0.51 (0.06) | 0.40 (0.04) | **0.47** (0.06) | 0.43 (0.04) | 0.39 (0.03) | **0.38** (0.05) | 0.39 (0.04) |
| **AALasso** | 0.56 (0.06) | **0.60** (0.11) | **0.57** (0.08) | 0.47 (0.05) | 0.45 (0.07) | **0.46** (0.05) | 0.45 (0.02) | 0.37 (0.06) | **0.40** (0.04) |

Table 3: Precision, Recall and F1-score in function of the dimension. We took a noise over the prior network with $\sigma_A = 0.1$ and $N = 40$ samples.

From Table 3, we see that in settings where $N << p$, AALasso remains better than the other standard methods regarding the F1 score. More of that, the higher the dimension, the higher the gap between LS+ALasso and AALasso regarding the F1 score (from less than 1.2 times better in dimension 40 to twice better in dimension 160). Thus, our method efficiently leverages prior knowledge, and it is of high interest in this kind of settings.

### 4.4.2 Influence of the prior network

In this section, we present results for various configurations of prior noises. Similar to the experiments in the previous section, for each configuration, we conducted 20 experiments, we took $N_{\text{iter}} = 10$, we utilized half of the training set and the parameters $\lambda$ and $\gamma$ were selected via cross-validation, minimizing the normalized Root Mean Square Error (nRMSE) over the second half. Figure 7 presents the results for Prediciton error and F1-score with $N = 40, 100, 140$ samples for varying noise levels. It is important to note that we are assessing the robustness of our method to prior matrix noise, where the noise specifically corresponds to $\mathbf{A}^{\text{prior}}$ noise and not to the noise of the VAR model. Additionally, since Lasso and LS+ALasso do not leverage prior knowledge, variations in this noise do not impact the results. The findings demonstrate that AALasso exhibits robustness to noise, displaying a comparable prediction error than the Lasso or LS+ALasso one (even better than LS+AALasso in few samples settings) and a consistently better F1-score for all tested configurations. Furthermore, these results indicate that our model effectively generalizes Adaptive Lasso (corresponding to the first iteration) and enables refinement of results over subsequent iterations.

Tables 4 and 5 provide further insights into the high F1-score achieved with AALasso for both $N = 40$ and $N = 140$ scenarios. The AALasso's behavior remains consistent across all tested noise levels. While the first iteration yields a good recall but limited precision, subsequent iterations lead to a significant increase in precision (an average gain of 0.15), especially in high noise level settings (gain of 0.2) while maintaining a good recall (an average loss of 0.05).

### 4.4.3 Comparison with other Causal Discovery algorithms

While we focused in this paper on learning Network Granger Causality, it is interesting to compare our method with other causal network discovery models. A recent survey Assaad et al. (2022) presents some state of the art methods to infer causality from time series. The main ones are based on the conditional independence framework Spirtes et al. (2000) which consists in running conditional independence tests to check whether a variable $X$ is a parent of another variable $Y$ conditioned to some other variables $Z_1, ..., Z_n$. Based on the PC algorithm Spirtes & Glymour (1991), in Runge et al. (2019), the authors introduced a new model (PCMCI) specific to causal discovery from times series, using what they call "the momentary conditional independence (MCI)" to remove false positive returned by the PC algorithm. Some variants of PCMCI were then designed, like PCMCI+ Runge (2020) which includes contemporaneous links and improves

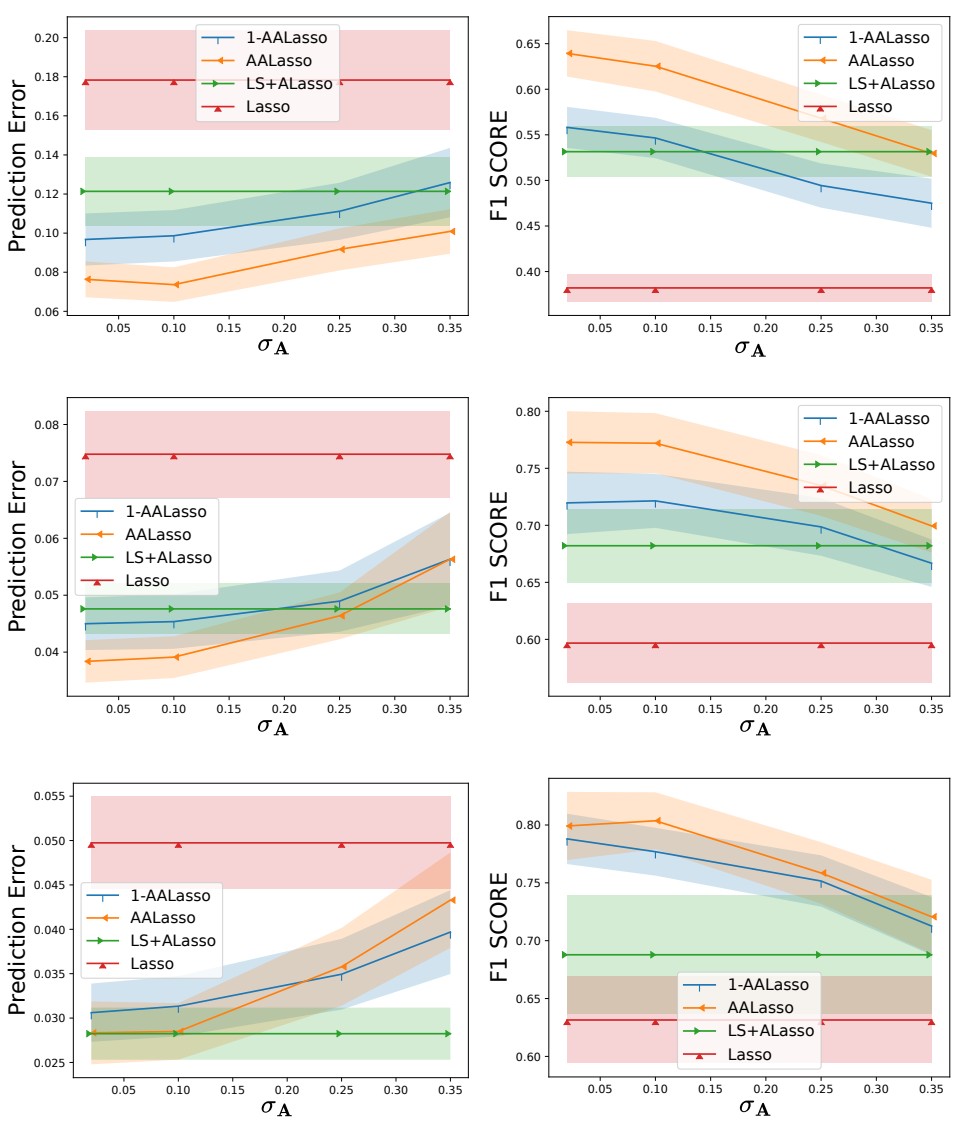

Figure 7: Prediction error and F1-score in function of the noise $\sigma_A$. (Top) $N = 40$ samples, (Middle) $N = 100$ samples, (Bottom) $N = 140$ samples.

the reliability of CI tests, or LPCMCI Gerhardus & Runge (2020) which is designed to address the issue of the presence of latent confounders.

Note that it is possible to include some prior knowledge when using methods based on conditional independence. These prior information can be expressed as: there is a causal link $i \mapsto j$, there is not any link between $i$ and $j$, $i$ is a leaf, $i$ is a root or $i$ is an ancestor of $j$. This kind of prior knowledge do not exactly match with the one used in this paper (the information is binary in this framework) but we tested PCMCI leveraging prior (denoted by Prior PCMCI) by encoding forbidding edges $(i, j)$ or $(j, i)$ when $\mathbf{A}_{i,j}^{\text{prior}} < \tau$ (here we tested $\tau \in \{0.2, 0.3\}$). All of these methods are implemented in `https://jakobrunge.github.io/tigramite/` and we used this package to perform the same experiments than the one in Section 4.4.1. We also add the results obtained with CGC-2SPR Yao et al. (2013), the method presented in Section 3.3 that leverage the same kind of prior knowledge than our method but using a $L2$ penalty.

| $\sigma_{\mathbf{A}}$ | 0.1 | | | 0.25 | | | 0.35 | | |
|---|---|---|---|---|---|---|---|---|---|
| **Metrics** | P | R | F1 | P | R | F1 | P | R | F1 |
| **Lasso** | 0.28 (0.04) | 0.61 (0.09) | 0.38 (0.04) | 0.28 (0.04) | 0.61 (0.09) | 0.38 (0.04) | 0.28 (0.04) | 0.61 (0.09) | 0.38 (0.04) |
| **LS+ALasso** | 0.54 (0.09) | 0.54 (0.11) | 0.53 (0.07) | **0.54** (0.09) | 0.54 (0.11) | 0.53 (0.07) | **0.54** (0.09) | 0.54 (0.11) | **0.53** (0.07) |
| **1-AALasso** | 0.44 (0.06) | **0.73** (0.1) | 0.55 (0.06) | 0.39 (0.06) | **0.71** (0.11) | 0.49 (0.07) | 0.37 (0.08) | **0.7** (0.11) | 0.47 (0.07) |
| **AALasso** | **0.58** (0.07) | 0.69 (0.11) | **0.63** (0.08) | 0.5 (0.07) | 0.67 (0.12) | **0.57** (0.07) | 0.45 (0.06) | 0.65 (0.12) | **0.53** (0.07) |

Table 4: Precision, Recall and F1-score in function of $\sigma_A$. We used $N = 40$ samples for training.

| $\sigma_{\mathbf{A}}$ | 0.1 | | | 0.25 | | | 0.35 | | |
|---|---|---|---|---|---|---|---|---|---|
| **Metrics** | P | R | F1 | P | R | F1 | P | R | F1 |
| **Lasso** | 0.54 (0.13) | 0.78 (0.08) | 0.63 (0.1) | 0.54 (0.13) | 0.78 (0.08) | 0.63 (0.1) | 0.54 (0.13) | 0.78 (0.08) | 0.63 (0.1) |
| **LS+ALasso** | 0.67 (0.25) | 0.77 (0.1) | 0.69 (0.14) | 0.67 (0.25) | 0.77 (0.1) | 0.69 (0.14) | 0.67 (0.25) | 0.77 (0.1) | 0.69 (0.14) |
| **1-AALasso** | 0.75 (0.05) | **0.81** (0.08) | 0.78 (0.06) | 0.71 (0.06) | **0.8** (0.08) | 0.75 (0.06) | 0.66 (0.07) | **0.78** (0.1) | 0.71 (0.07) |
| **AALasso** | **0.82** (0.06) | 0.79 (0.09) | **0.8** (0.07) | **0.76** (0.06) | 0.76 (0.1) | **0.76** (0.07) | **0.71** (0.09) | 0.74 (0.11) | **0.72** (0.09) |

Table 5: Precision, Recall and F1-score in function of $\sigma_A$. We used $N = 140$ samples for training.

| **N Samples** | 40 | | | 100 | | | 180 | | |
|---|---|---|---|---|---|---|---|---|---|
| **Metrics** | P | R | F1 | P | R | F1 | P | R | F1 |
| **Lasso** | 0.28 (0.04) | 0.61 (0.09) | 0.38 (0.04) | 0.51 (0.1) | 0.75 (0.08) | 0.6 (0.09) | 0.57 (0.08) | 0.83 (0.06) | 0.68 (0.07) |
| **LS+ALasso** | 0.54 (0.09) | 0.54 (0.11) | 0.53 (0.07) | 0.69 (0.1) | 0.68 (0.1) | 0.68 (0.09) | 0.64 (0.08) | 0.83 (0.06) | 0.72 (0.07) |
| **CGC-2SPR** | 0.45 (0.22) | 0.42 (0.16) | 0.36 (0.07) | 0.66 (0.22) | 0.57 (0.11) | 0.58 (0.10) | 0.84 (0.12) | 0.64 (0.10) | 0.72 (0.06) |
| **PCMCI+** | **0.89** (0.08) | 0.38 (0.07) | 0.53 (0.07) | **0.92** (0.05) | 0.56 (0.06) | 0.69 (0.05) | **0.92** (0.04) | 0.67 (0.05) | 0.77 (0.04) |
| **LPCMCI** | 0.64 (0.08) | 0.5 (0.06) | 0.56 (0.06) | 0.64 (0.07) | 0.63 (0.06) | 0.63 (0.06) | 0.64 (0.08) | 0.72 (0.06) | 0.68 (0.06) |
| **Prior PCMCI+** | 0.68 (0.07) | 0.5 (0.06) | 0.58 (0.05) | 0.7 (0.06) | 0.64 (0.06) | 0.66 (0.05) | 0.71 (0.05) | 0.74 (0.05) | 0.72 (0.04) |
| **1-AALasso** | 0.44 (0.06) | **0.73** (0.1) | 0.55 (0.06) | 0.65 (0.06) | **0.82** (0.09) | 0.72 (0.06) | 0.83 (0.06) | **0.84** (0.06) | 0.83 (0.05) |
| **AALasso** | 0.55 (0.07) | 0.71 (0.11) | **0.62** (0.07) | 0.76 (0.06) | 0.79 (0.1) | **0.77** (0.07) | 0.87 (0.05) | 0.81 (0.07) | **0.84** (0.05) |

Table 6: Precision, Recall and F1-score for all causal discovery methods tested, $p = 40$, $\sigma_A = 0.1$.

Compared to methods that do not leverage prior knowledge (Lasso, PCMCI and its variants), Table 6 show that AALasso allows for better network reconstruction, highlighting its ability to efficiently leverage prior information. Moreover, AALasso remains superior to Prior PCMCI+, showing that the way we introduce the prior information in our model is relevant and efficient. In details, we remark that PCMCI+ allows for a very good Precision for the retrieved causal links (from 0.89 to 0.92), but the Recall remains very low (from 0.38 to 0.67). This method achieves the best Precision but the worst Recall, leading to a F1-score lower than the one of AALasso.

### 4.4.4  Empirical time complexity

Concerning time complexity, as discussed theoretically in Section 3.5.2, we observed that the runtime of AALasso is approximately proportional to $N_{\text{iter}}$ times that of a Lasso estimator. Therefore, it is useful to analyze the convergence rate of our method to find a balance between precision and computational efficiency. Figure 8 shows the behavior of the Prediction error and F1-score averaged on all experiments conducted to visualize the convergence over the successive iterations. The convergence seems to be achieved for $N_{\text{iter}} = 10$ in average (see Appendix B.1 for more experiments), and that motivated the choice of $N_{\text{iter}}$ for comparing the performances of the algorithms. Then we can compare runtimes of Lasso, LS+ALasso and AALasso with $N_{\text{iter}} = 10$ iterations. To well understand the gain and the tradeoff between runtime and performances,

we plot the F1 score in function of the runtimes for $N_{\text{iter}} \in \{0, 5, 10, 15\}$ with $N = 40$ samples and $\sigma_A = 0.1$ in Figure 9 and for other scenarios in Appendix B.5.

We remark that AALasso with 5 iterations is a good trade off to optimize both the runtime and the F1-score.

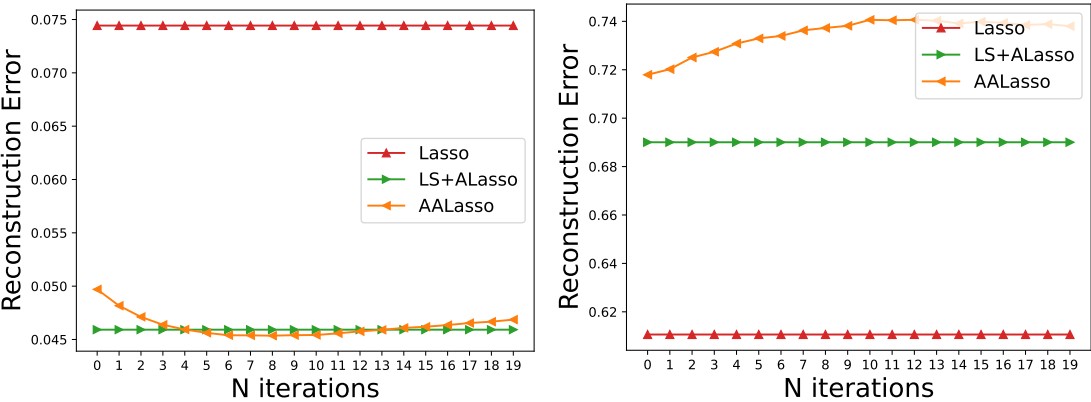

Figure 8: Prediction error and F1-score in function of the number of iterations used for training using synthetic data with $\sigma_A \in \{0.02, 0.1, 0.25, 0.35\}$ and $N \in \{40, 100, 140, 180, 250\}$.

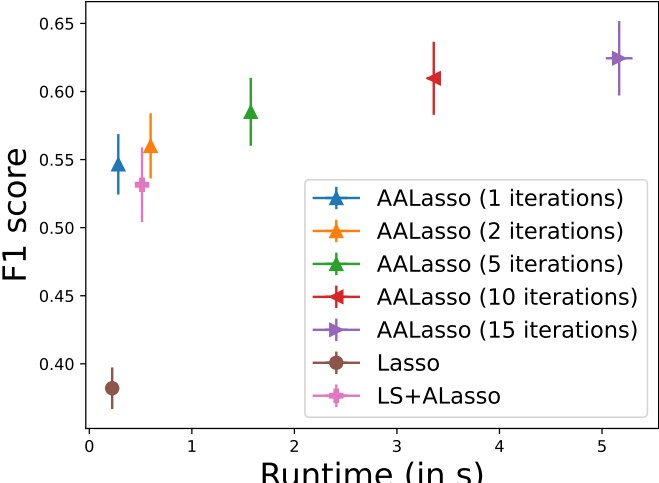

Figure 9: F1 score in function of runtimes for Lasso, LS+ALasso and AALasso for $N_{\text{iter}} \in \{0, 5, 10, 15\}$ with $N = 40$ samples, $\sigma_A = 0.1$.

## 4.5 Experiments on real-world data

### 4.5.1 HPN DREAM8 Challenge

For this dataset, the choice of $\mathbf{A}^{\text{prior}}$ is detailed in Section 4.3.2 and we compare the results with Lasso and Least Squares methods (note that we replace the LS+ALasso method here because the results were better when fitting the Least Squares estimator).

The hyperparameters are selected using data for the cells BT549, MCF7, and UACC812 and the test set contains the data for all pairs (ligand, inhibitor) for the cell BT20. We use two common metrics to compare the performances of the algorithms : the overall accuracy, measuring the relevance of variable selection, and

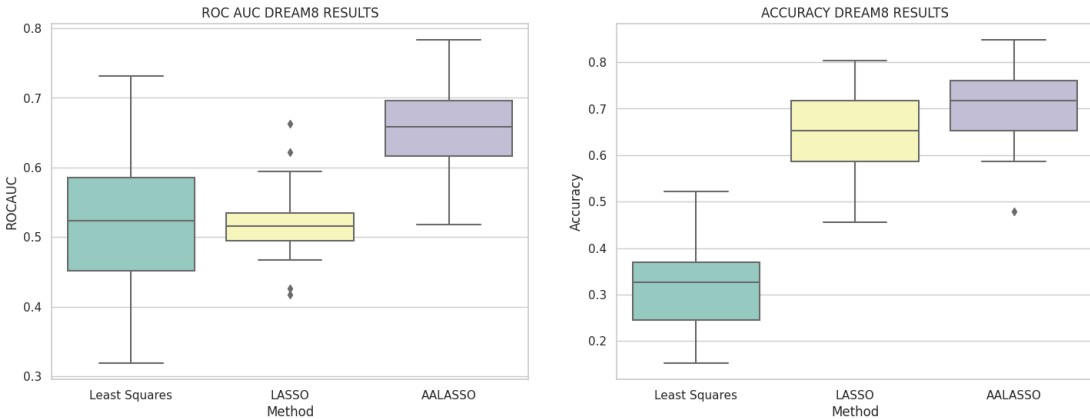

Figure 10: Results obtained on the DREAM8 dataset

the Receiver Operating Characteristic Area Under the Curve (ROCAUC), allowing to design a threshold-independent score as mentioned in (Bradley, 1997) and measuring the ability of a model to discriminate the two classes. Formally, let $(C_{i,j})_{1 \leq i,j \leq p} \in \mathbb{R}^{p \times p}$ the parameters computed and $(Y_{i,j})_{1 \leq i,j \leq p} \in \{0,1\}^{p \times p}$ the ground truth values (causality or not), then

- The accuracy is computed by : $Acc = \dfrac{\sum\limits_{1 \leq i,j \leq p} T_\tau(C_{i,j})Y_{i,j}}{p^2}$, where $T_\tau : x \mapsto \begin{cases} 1 \text{ if } |x| > \tau \\ 0 \text{ otherwise} \end{cases}$, taking $\tau = 0.05$. This metrics measures the relevance of variable selection.

- The ROCAUC is computed by $\mathrm{ROCAUC} = \int_0^1 \mathrm{TPR}(\mathrm{FPR}^{-1}(t)) \, dt$ where $\mathrm{TPR}(t)$ stands for True Positive Rate (sensitivity), and $\mathrm{FPR}(t)$ stands for False Positive Rate (1 - specificity) with a threshold $t$.

As shown in Figure 10 our method AALasso demonstrates better performance when compared to traditional estimators such as Lasso and Ordinary Least Squares (OLS) regarding both accuracy and ROCAUC. Thus, by incorporating the adjacency matrix derived from the Pathway Commons database as a prior network, AALasso effectively harnessed valuable prior knowledge about protein interactions. Our results showcase that AALasso outperform the baseline methods, emphasizing the significance of considering such prior information in causality inference tasks. The Figure 11 presents an example of graph learned, and we remark that the Lasso estimator explain all the variables only with a few number of variables (presence of columns in the matrix). On the contrary, the AALasso estimates are more homogeneous, following a directed version of the prior structure.

### 4.5.2 Molene Dataset

**VAR(1) model**  For this dataset, we train the models with 80 points, still selecting $\lambda$ and $\gamma$ by cross-validation. Figure (12) compares the resulting graphs of Granger causalities using our method AALasso and Lasso. We observe that the graph resulting of the AALasso is sparse while remaining connected and allows a good visualization of the physical process. Moreover, contrarily to the Lasso one, it is consistent with the Euclidean structure, confirming that the algorithm leverages the given prior matrix. This simple example encourages the using of AALasso to learn Granger causality for dynamic or physic system by taking into account the physics of the problem.

**VAR(3) model**  For this example, we applied the generalization of our model with the order $d = 3$. We computed the same matrix $\mathbf{A}^{\mathrm{prior}}$ than for $d = 1$ (cf Section 4.3.3) and we then considered the generalized

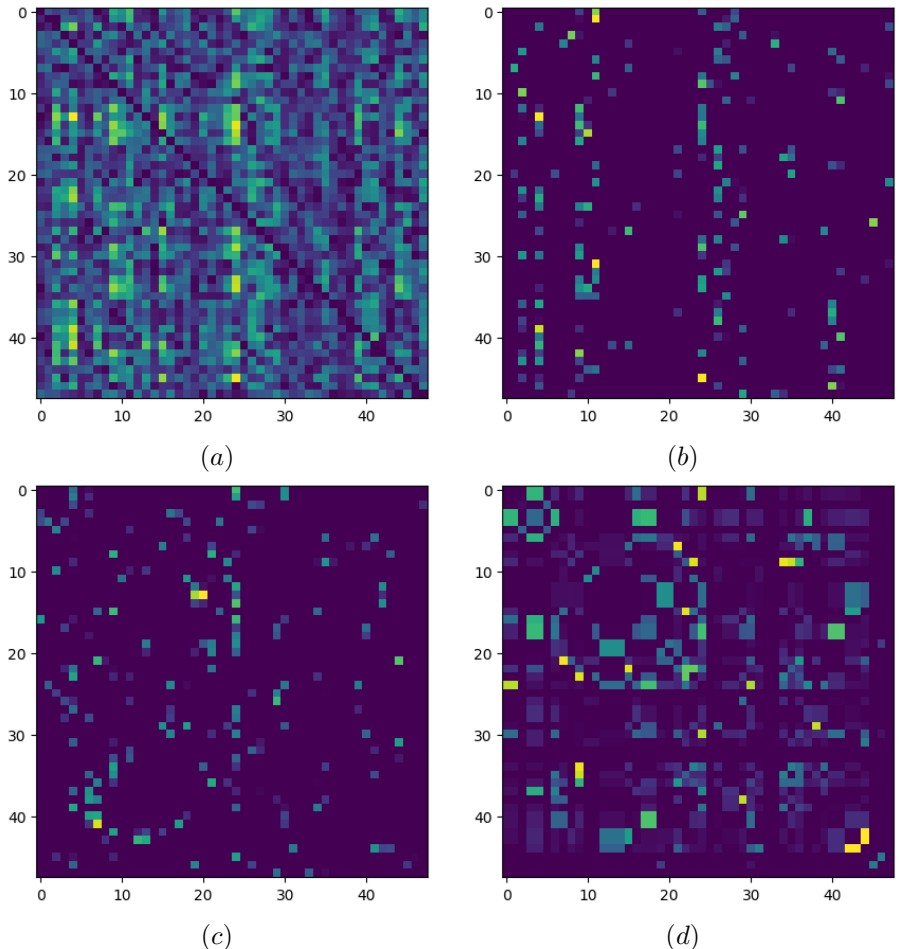

Figure 11: Example of graphs learned. (a) Least Squares method. (b) Lasso method. (c) AALasso. (d) Prior network.

prior matrix given by

$$\overline{\mathbf{A}^{\text{prior}}} = \begin{bmatrix} \mathbf{A}^{\text{prior}} & \mathbf{A}^{\text{prior}} & \cdots & \mathbf{A}^{\text{prior}} \\ 1 & 0 & \cdots & 0 \\ \vdots & \ddots & \ddots & \vdots \\ 0 & \cdots & 1 & 0 \end{bmatrix}.$$

The comparison between the LS+ALasso and the AALasso estimates for $d = 3$ is presented if Figure 13. Like for the case $d = 1$, the AALasso graph is sparser than the LS+ALasso one, and allows a better visualization of the physical process. Moreover, we remark that AALasso explains the main part of the signal only by using the first order (only 3 and 1 edges for order 2 and 3) which is consistent with a diffusion process. Finally, the size of the edges for AALasso seems to be proportional to the order (edges for order 2 and 3 are longer than the ones for order 1) which is again consistent with the physics (information take more time to travel longer distances).

## 5 Discussion and conclusion

In conclusion, this paper has introduced a novel method designed to efficiently learn Granger causalities in settings with limited samples. Our approach stands out by effectively incorporating prior knowledge through the utilization of a noisy adjacency matrix. We demonstrated the convergence of our algorithm AALasso and

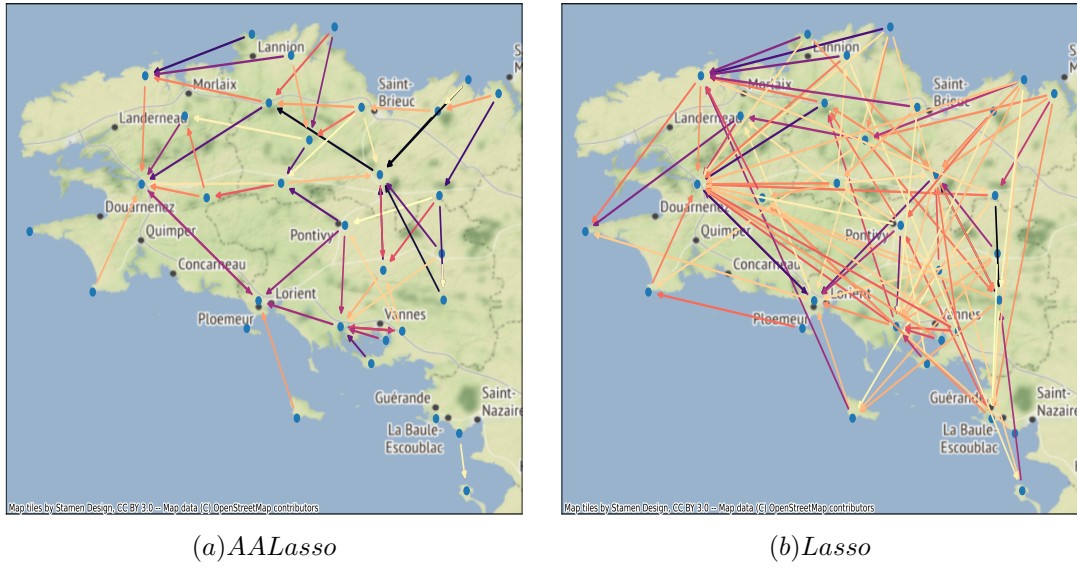

(a)AALasso                    (b)Lasso

Figure 12: C Results on the Molène dataset for (a) AALasso and (b) Lasso. Darker colors indicate larger weight.

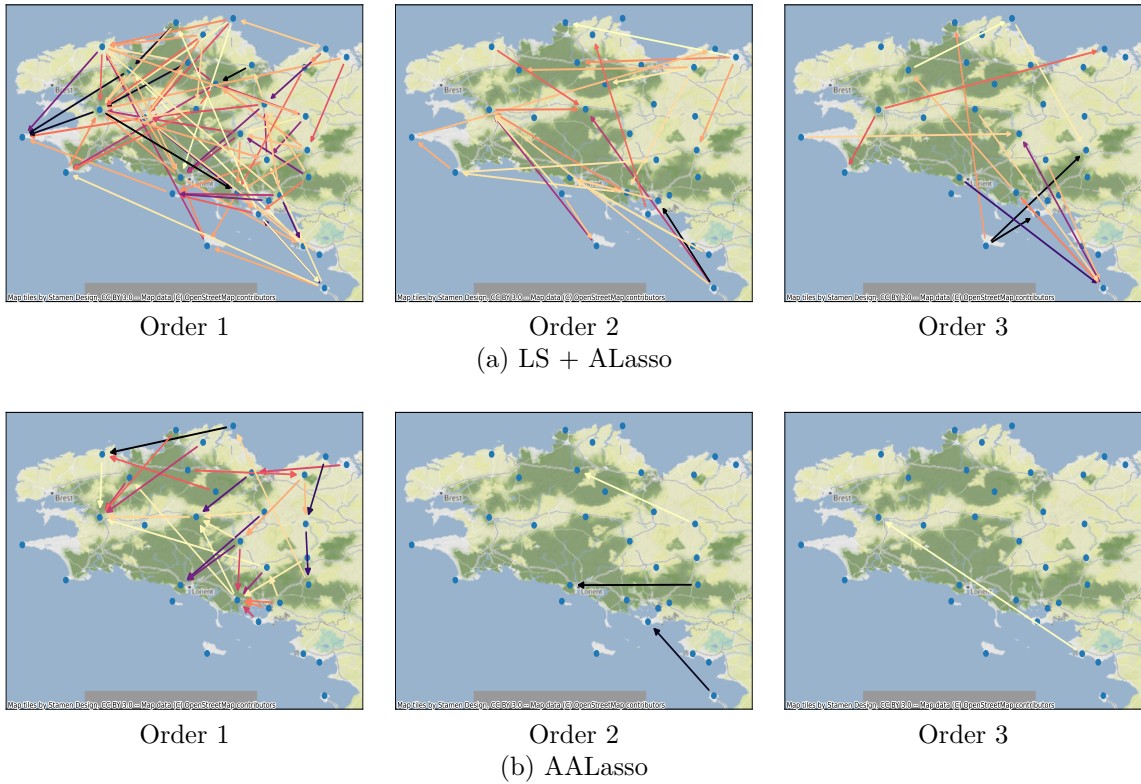

Order 1                Order 2                Order 3
(a) LS + ALasso

Order 1                Order 2                Order 3
(b) AALasso

Figure 13: C Results on the Molène dataset with $d = 3$ for (a) LS + ALasso and (b) AALasso. Darker colors indicate larger weight.

showed that the time complexity was in the same order of magnitude as that of the Lasso. To empirically validate our method and demonstrate its efficacy, we conducted a series of experiments. We selected a variety of datasets, including synthetic data and real-world examples like the Breast Cancer Network and the Molène

Dataset. In these experiments, we employed rigorous evaluation metrics to assess the performance of our method across different scenarios, showcasing its versatility and applicability. Thus the incorporation of prior information has proven instrumental in achieving superior accuracy and robustness when compared to classical algorithms in this domain.

While our method allows to incorporate prior knowledge in the learning process, it could be interesting to add structure to the learned graph and go beyond sparsity. Indeed, the framework we have presented here can be readily extended to learn graphs with specific structural constraints, such as spectral and adjacency constraints, similar to those outlined in (Kumar et al., 2020). This flexibility arises from the fact that our model operates with a symmetric matrix containing positive values, making it amenable to a range of applications that necessitate tailored graph structures.

Finally, it could be interesting to study whether this framework could be used to learn time-varying graphs of Granger causality (cf (Gao & Yang, 2022)), for example by considering the graph learned at time $t$ as a prior for the graph at time $t + 1$.

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

## A   Proofs

### A.1   Statistical model

*Proof of* (13). The MAP of the model (4) is given by:

$$\widehat{\mathbf{A}}, \widehat{\mathbf{C}} = \underset{\mathbf{A}, \mathbf{C}}{\arg\max} \; L(\mathbf{A}, \mathbf{C} \mid \{X_i[1:t]\}_{i=1}^N, \mathbf{A}^{\mathrm{prior}}) \tag{17}$$

$$\text{subject to} \; \; \mathbf{A} \geq 0 \tag{18}$$

where $L(\cdot)$ is the likelihood function. Using Bayes formula, ones have :

$$
\begin{aligned}
L(A^*, C^* \mid X^{1:t}, \mathbf{A}^{\mathrm{prior}}) &:= \mathbb{P}(A^*, C^* \mid X^{1:t}, \mathbf{A}^{\mathrm{prior}}) \\
&= \mathbb{P}(C^* \mid X^{1:t}, \mathbf{A}^{\mathrm{prior}}) \times \mathbb{P}(A^* \mid C^*, X^{1:t}, \mathbf{A}^{\mathrm{prior}}) \\
&= \mathbb{P}(C^* \mid X^{1:t}, \mathbf{A}^{\mathrm{prior}}) \times \mathbb{P}(A^* \mid C^*, \mathbf{A}^{\mathrm{prior}}) \\
&= \frac{\mathbb{P}(X^{1:t} \mid C^*) \times \mathbb{P}(C^* \mid \mathbf{A}^{\mathrm{prior}})}{\mathbb{P}(X^{1:t} \mid \mathbf{A}^{\mathrm{prior}})} \times \mathbb{P}(A^* \mid C^*, \mathbf{A}^{\mathrm{prior}}) \\
&= \frac{\mathbb{P}(X^{1:t} \mid C^*)}{\mathbb{P}(X^{1:t} \mid \mathbf{A}^{\mathrm{prior}})} \times \big(\mathbb{P}(C^* \mid \mathbf{A}^{\mathrm{prior}}) \times \mathbb{P}(A^* \mid C^*, \mathbf{A}^{\mathrm{prior}})\big)
\end{aligned}
$$

Then:

$$
\begin{aligned}
\mathbb{P}(C^* \mid \mathbf{A}^{\mathrm{prior}}) \times \mathbb{P}(A^* \mid C^*, \mathbf{A}^{\mathrm{prior}}) &= \mathbb{P}(C^* \mid A^*, \mathbf{A}^{\mathrm{prior}}) \times \mathbb{P}(A^* \mid \mathbf{A}^{\mathrm{prior}}) \\
&= \mathbb{P}(C^* \mid A^*) \times \mathbb{P}(A^* \mid \mathbf{A}^{\mathrm{prior}}) \\
&= \mathbb{P}(C^* \mid A^*) \times \mathbb{P}(\mathbf{A}^{\mathrm{prior}} \mid A^*) \frac{\mathbb{P}(A^*)}{\mathbb{P}(\mathbf{A}^{\mathrm{prior}})}
\end{aligned}
$$

Assuming that $A^*$ follows the improper distribution $\mathbf{1}_{S_p(\mathbb{R})}$, the MAP is finally given by:

$$
\begin{aligned}
&\max_{A^*, C^*} \mathbb{P}(X^{1:t} \mid C^*) \times \mathbb{P}(C^* \mid A^*) \times \mathbb{P}(\mathbf{A}^{\mathrm{prior}} \mid A^*) \\
&= \max_{A^*, C^*} \mathcal{N}(X^t; C^* X^{t-1}, \sigma_X^2 Id) \times \prod_{i,j} Laplace(C_{i,j}^*; 0, A_{i,j}^*) \times \prod_{i,j} \mathcal{N}(\mathbf{A}_{i,j}^{\mathrm{prior}}; A_{i,j}^*, \sigma^2)
\end{aligned}
$$

Applying the log function to the previous expression concludes the proof. $\qquad\square$

### A.2   Proof of Theorem 12

**Definition 14** (Regular function)**.** *We say that $f$ is regular at $z \in dom f$ if $f'(z; d) \geq 0$, $\forall d = (d_1, ..., d_p)$, such that $f'(z; (0, ..., d_k, ..., 0)) \geq 0, k = 1, ..., p$.*

**Remark 15.** *If $f$ is differentiable then $f'(x; d) = \nabla f(x)^T d$. So, if $f'(z; (0, ..., d_k, ..., 0)) \geq 0 \;\; \forall k = 1, ..., p$, we have that:*

$$f'(z; (d_1, ..., d_k, ..., d_p)) = \nabla f(x)^T (d_1, ..., d_k, ..., d_p) = \sum_{k=1}^p \nabla f(x)^T (0, ..., d_k, ..., 0) \geq 0. \tag{19}$$

*Thus a differentiable function is regular.*

**Definition 16** (Stationary points)**.** *We say that $z$ is a **stationary point** of $f$ if $z \in dom f$ and $f'(z; d) \geq 0$, $\forall d$.*
*We say that $z$ is a **coordinatewise minimum** point of $f$ if $z \in dom f$ and $f(z + (0, ..., d_k, ..., 0)) \geq f(z)$, $\forall d_k \in \mathbb{R}^{n_k}$, for all $k = 1, ..., N$.*

**Remark 17.** *If $z$ is a coordinatewise minimum point of $f$, $z$ is a stationary point of $f$ whenever $f$ is regular at $z$.*

Using this framework, the following theorem holds.

**Theorem 18** (Theorem 4.1 in (Tseng, 2001)). *Assume that the level set $\mathcal{L}_X = \{x \mid f(x) \leq f(x_0)\}$ is compact (where $x_0 \in \mathbb{R}$) and that $f$ is continuous on $\mathcal{L}_X$. Then, the sequence $\{x^r\}_{r=1,2,...}$ generated by the Block Coordinate Descent method is defined and bounded. Moreover, the following statements hold: If $f(x_1,...,X^{(n)})$ has at most one minimum in $x_k$ for $k \in \{2,...,N-1\}$, then every cluster point $z$ of $\{x^r\}_{r \equiv N-1 \mod N}$ is a coordinatewise minimum point of $f$. In addition, if $f$ is regular at $z$, then $z$ is a stationary point of $f$.*

Actually the case $N = 2$ is very simple and we do not need any supplementary assumptions that the continuity of $f$ and the compactness of $\mathcal{L}_X$ to prove the convergence of the algorithm. Indeed, since $\{2,...,N-1\} = $ for the case $N = 2$, the point (3) can be directly applied without satisfying any assumptions about the number of minimum so ones obtain directly the following corollary.

**Corollary 19.** *Assume that the level set $\mathcal{L}_X$ is compact, that $f$ is continuous on $\mathcal{L}_X$ and that $N = 2$. Then, the sequence $\{x^r\}_{r=1,2,...}$ generated by the BCD method is defined and bounded. Moreover, every cluster point $z$ of $\{x^r\}_{r \equiv N-1 \mod N}$ is a coordinatewise minimum point of $f$. In addition, if $f$ is regular at $z$, then $z$ is a stationary point of $f$.*

A simplified version of the proof provided in (Tseng, 2001) for the case $N = 2$ can be found in Appendix (A.2).

Now we need to prove that the function $f$ of our model (the MAP) satisfies the conditions of the previous theorem.

**Proposition 20.** *The function $f : \mathbb{R}^{p^2} \times \mathbb{R}_+^{*\,p^2} \to \mathbb{R}$ defined by :*

$$f(\mathbf{C}, \mathbf{A}) = \frac{1}{N} \sum_{n=1}^{N} ||X^{(n)}[t] - \mathbf{C}X^{(n)}[t-1]||_2^2 + \lambda \sum_{i<j} \frac{|\mathbf{C}_{i,j}| + |\mathbf{C}_{j,i}|}{\mathbf{A}_{i,j}} + \lambda \sum_{i<j} \log(2\mathbf{A}_{i,j}) + \gamma||\mathbf{A} - \mathbf{A}^{prior}||_F^2 \quad (20)$$

*is regular.*

**Proposition 21.** *The function $f$ defined in (20) but constrained on $\mathbb{R}^{p^2} \times [\epsilon, +\infty]^{p^2}$ with $\epsilon > 0$ satisfies assumptions in (19).*

Finally, using 20 and 21 we show that our objective function in (7) verifies the assumptions in (19), and applying the Theorem, we obtain the following result.

*Proof of 19.* Let $\{x^r\}_{r=1,2,...}$ the sequence generated with the BCD algorithm. By definition of the algorithm ones have that $f(x^{r+1}) \leq f(x^r)$ for all $r$ and $x^{r+1} \in \mathcal{L}_X$ for all $r$. Since $\mathcal{L}_X$ is compact, $\{x^r\}_{r \in \mathcal{R}}$ converges towards $z = z^1$. In the same way, we can assume w.l.o.g that $\{x^{r+1}\}_{r \in \mathcal{R}}$ converges towards $z^2$ (taking a sub-sequence).

First, $\{f(x^r)\}_{r \in \mathcal{R}}$ is decreasing (and bounded) so it converges and ones have that :

$$f(x^0) \geq \lim_{r \in \mathcal{R} \to +\infty} f(x^r) = f(z) = f(z^1) = f(z^2).$$

Now, we assume that for every $r \in \mathcal{R}$, $x^r = \underset{x}{argmin} f(x, x_2^{r-1})$, e.g for all $r$:

$$f(x^r) \leq f(x^r + (d_1, 0)), \quad \forall d_1$$
$$f(x^{r+1}) \leq f(x^{r+1} + (0, d_1)), \quad \forall d_2$$
$$x_1^r = x_1^{r+1} \quad \text{where} \quad x^r = (x_1^r, x_2^r)$$

Since $f$ is continuous on $\mathcal{L}_X$, we get :

$$f(z^1) \leq f(z^1 + (d_1, 0)), \quad \forall d_1$$
$$f(z^2) \leq f(z^2 + (0, d_2)), \quad \forall d_2$$
$$z_1^2 = z_1^1$$

Then, for all $d_2$ :

$$\begin{aligned}
f(z^1) &= f(z^2) \\
&\leq f((z_1^2, z_2^2) + (0, d_2)) \\
&= f((z_1^1, z_2^2) + (0, d_2)) \\
&= f((z_1^1, z_2^1) + (0, z_2^2 - z_2^1) + (0, d_2)) \\
&= f(z^1 + (0, \tilde{d}_2))
\end{aligned}$$

Since $z^1 = z$, we proved that for all $d_1, d_2$ :

$$f(z) \leq f(z + (d_1, 0)), \quad \forall d_1$$
$$f(z) \leq f(z + (0, d_2)), \quad \forall d_2$$

so $z$ is a componentwise minimum of $f$.

Finally, if $f$ is regular, the previous inequalities become :

$$f'(z; (d_1, 0)) \geq 0, \quad \forall d_1$$
$$f'(z; (0, d_2)) \geq 0, \quad \forall d_2$$

and by definitions $z$ is a stationary point of $f$. $\qquad\square$

*Proof of 20.* The only points where $f$ is not differentiable are $\{(\mathbf{C}, \mathbf{A}) \mid \exists i, j \; ; \; \mathbf{C}_{i,j} = 0\}$ because of the absolute value. Let's write :

$$f(\mathbf{C}, \mathbf{A}) = g(\mathbf{C}, \mathbf{A}) + \lambda \sum_{i<j} h_{i,j}(\mathbf{C}, \mathbf{A}) + l(\mathbf{C}, \mathbf{A})$$

where

$$g(\mathbf{C}, \mathbf{A}) = \frac{1}{N} \sum_{n=1}^{N} ||X^{(n)}[t] - \mathbf{C}X^{(n)}[t-1]||_2^2$$

$$h_{i,j}(\mathbf{C}, \mathbf{A}) = \frac{|\mathbf{C}_{i,j}| + |\mathbf{C}_{j,i}|}{\mathbf{A}_{i,j}}$$

$$l(\mathbf{C}, W) = \lambda \sum_{i<j} \log(2\mathbf{A}_{i,j}) + \gamma ||\mathbf{A} - \mathbf{A}^{\text{prior}}||_F^2.$$

$g$ and $l$ are differentiable so we have for all $(\mathbf{C}, \mathbf{A}) \in \mathbb{R}^{p^2} \times \mathbb{R}_+^{*\,p^2}$, for all $(D_C, D_\mathbf{A}) \in \mathbb{R}^{p^2} \times \mathbb{R}^{p^2}$ such that $(\mathbf{C} + D_C, \mathbf{A} + D_\mathbf{A}) \in dom f$ :

$$g'((\mathbf{C}, \mathbf{A}); (D_C, D_\mathbf{A})) = \sum_{i,j} g'((\mathbf{C}, \mathbf{A}); (D_C^{(i,j)}, 0)) + g'((\mathbf{C}, \mathbf{A}); (0, D_Z^{(i,j)}))$$

$$l'((\mathbf{C}, \mathbf{A}); (D_C, D_{\mathbf{A}})) = \sum_{i,j} l'((\mathbf{C}, \mathbf{A}); (D_C^{(i,j)}, 0)) + l'((\mathbf{C}, \mathbf{A}); (0, D_{\mathbf{A}}^{(i,j)}))$$

where $D_C^{(i,j)}$ is the matrix with zero values everywhere except the coefficient $(i, j)$ which is equal to $D_C^{(}i, j)$. If $\mathbf{C}_{i,j} \neq 0$ and $\mathbf{C}_{j,i} \neq 0$, then $h_{i,j}$ is differentiable in $(\mathbf{C}, \mathbf{A})$ so we have the same result. Otherwise, we need to compute the lower directional derivative of $h_{i,j}$ in $(\mathbf{C}, W)$ with $\mathbf{C}_{i,j} = 0$ or $\mathbf{C}_{j,i} = 0$.
Ones can compute that:

- $h'_{i,j}((\mathbf{C}, \mathbf{A}); (D_C, D_{\mathbf{A}})) = \frac{|D_C^{(i,j)}| + |D_C j,i|}{\mathbf{A}_{i,j}}$ if $\mathbf{C}_{i,j} = 0$ and $\mathbf{C}_{j,i} = 0$

- $h'_{i,j}((\mathbf{C}, \mathbf{A}); (D_C, D_{\mathbf{A}})) = \frac{|D_C^{(i,j)}|}{\mathbf{A}_{i,j}} + \frac{\text{sign}(C_{j,i}) D_C^{(j,i)}}{\mathbf{A}_{i,j}} - \frac{D_{\mathbf{A}}^{(i,j)} |C_{j,i}|}{\mathbf{A}_{i,j}^2}$ if $\mathbf{C}_{i,j} = 0$ and $\mathbf{C}_{i,j} \neq 0$ and we can do the same for the last case.

Thus we still have that :

$$h'_{i,j}((\mathbf{C}, \mathbf{A}); (D_C, D_{\mathbf{A}})) = \sum_{i,j} h'_{i,j}((\mathbf{C}, \mathbf{A}); (D_C^{(i,j)}, 0)) + h'_{i,j}((\mathbf{C}, \mathbf{A}); (0, D_{\mathbf{A}}^{(i,j)})).$$

Finally, by definition of regular function, ones have that $f$ is regular at all $(\mathbf{C}, \mathbf{A}) \in \mathbb{R}^{p^2} \times \mathbb{R}_+^{* p^2}$. □

*Proof of 21.* First it is clear that $f$ is continuous on $\mathcal{L}_X$.
Let's consider again the decomposition :

$$f(\mathbf{C}, \mathbf{A}) = g(\mathbf{C}) + \lambda \sum_{i,j} h_{i,j}(\mathbf{C}, \mathbf{A}) + l(\mathbf{A})$$

where

$$g(\mathbf{C}) = \frac{1}{N} \sum_{n=1}^{N} ||X^{(n)}[t] - \mathbf{C} X^{(n)}[t-1]||_2^2$$

$$h_{i,j}(\mathbf{C}, \mathbf{A}) = \frac{|\mathbf{C}_{i,j}| + |\mathbf{C}_{j,i}|}{\mathbf{A}_{i,j}}$$

$$l(|\mathbf{C}_{i,j}|, \mathbf{A}) = \lambda \sum_{i<j} \log(2\mathbf{A}_{i,j}) + \gamma ||\mathbf{A} - \mathbf{A}^{\text{prior}}||_F^2.$$

It is clear that $\lim_{||\mathbf{C}|| \to +\infty} g(\mathbf{C}) = +\infty$ and $\lim_{||\mathbf{A}|| \to +\infty} l(\mathbf{A}) = +\infty$.
Then, since $h_{i,j}(\mathbf{C}, \mathbf{A}) \geq 0$ for all $\mathbf{C}, \mathbf{A}$ for all $i, j$, we have that :

$$\lim_{||(C, \mathbf{A})|| \to +\infty} f(\mathbf{C}, \mathbf{A}) = +\infty. \tag{21}$$

We proved that $f$ was coercive, it follows that $\mathcal{L}_X$ is bounded.
Moreover, $\lim_{\mathbf{A}_{i,j} \to 0} l(\mathbf{A}) = +\infty$, for $i, j \in [[1, p]]$. Additionally, $f$ is continuous so $f^{-1}(] - \infty, f(x^{(0)})])$ is closed and finally $\mathcal{L}_X$ **is compact**.

□

**Proposition 22.** *The function $f$ defined in (20) is not convex.*

*Proof of 22.* The function 20 is a function of $(\mathbf{C}, (\mathbf{A}_{i,j})_{1 \leq i,j \leq p})$, so it is sufficient to prove that it is not convex in $\mathbf{A}_{i,j}$ for fixed $i$ and $j$ and for a fixed value of $\mathbf{C}$. The second derivative wrt $\mathbf{A}_{i,j}$ is :

$$\partial_{i,j}^2 f(\mathbf{C}, \mathbf{A}) = -\frac{\lambda}{\mathbf{A}_{i,j}^2} + 2 \frac{|C_{i,j}| + |C_{j,i}|}{\mathbf{A}_{i,j}^3} + 2\gamma \tag{22}$$

so it has the same sign than the degree 3 polynomial:

$$-\lambda \mathbf{A}_{i,j} + 2(|C_{i,j}| + |C_{j,i}|) + 2\gamma \mathbf{A}_{i,j}^3 \tag{23}$$

Then, the minimum of this polynomial on $[0, +\infty]$ is reached in $\sqrt{\frac{\lambda}{6\gamma}}$ and take the value $-\frac{2\lambda^{3/2}}{3\sqrt{6\gamma}} + 2(|C_{i,j}| + |C_{j,i}|)$ which can be negative for small values of $|C_{i,j}| + |C_{j,i}|$.

Thus the second derivative can reach negative value with certain value of $\mathbf{C}_{i,j}$ and $\mathbf{C}_{j,i}$. $\qquad\square$

### A.3 Time complexity

**Lemma 23** (**C** update time complexity)**.** *Let $\mathcal{C}_{Lasso}(p, N)$ the time cost for training a Lasso estimator to fit VAR(1) parameters in dimension $p$ with $N$ samples, then the time complexity of a **C** update is in $O\left(p \times N + \mathcal{C}_{Lasso}(p, N)\right)$.*

*Proof of 23.* The $C$ update is done solving an Adaptive Lasso problem, so recalling that an Adaptive Lasso problem can be written as a Lasso problem in $p \times N$ multiplications and that we solve $p$ Adaptive Lasso problems (cf 3.4.1), the time complexity of this step is in $O\left(p^2 \times N + \mathcal{C}_{\mathrm{Lasso}}(p, N)\right)$. $\qquad\square$

**Lemma 24** (**A** update time complexity)**.** *The time complexity of a **A** update is in $O\left(p^2\right)$.*

*Proof of 24.* The **A** update is done by computing in $O(1)$ the closed form given in 3.4.2 for each value $\mathbf{A}_{i,j}$ so this update is in $O(p^2)$. $\qquad\square$

In order to completely express the time complexity, we need to compute $\mathcal{C}_{\mathrm{Lasso}}(p, N)$. Note that when using gradient descent based methods to solve an optimization problem, whereas a convergence rate analysis can be conducted (cf (Zhao & Huo, 2023)), the time complexity depends on the stop criterion of the algorithm. Thus we conduct the time complexity analysis assuming that ADMM is utilized with a fixed number of iterations $N_{\mathrm{ADMM}}$.

**Lemma 25** (ADMM complexity for Lasso)**.** *The time complexity of the ADMM algorithm (with a fixed number of steps) to solve one Lasso problem in dimension $p$ is $O(p^3 + N \times p^2)$.*

*Proof of 25.* The updated formula of the ADMM given in 3.4.1 require to multiply a $p \times N$ matrix with a $N \times p$ matrix which is in $O(N \times p^2)$ and to inverse a $p \times p$ matrix which is in $O(p^3)$. $\qquad\square$

**Theorem 26.** *Let $\mathcal{C}_{AALasso}(p, N)$ the time cost for training a AALasso estimator to fit VAR(1) parameters in dimension $p$ with $N$ samples, then:*

$$\mathcal{C}_{AALasso}(p, N) \underset{p,N}{=} O(\mathcal{C}_{Lasso}(p, N)) \underset{p,N}{=} O\left(p^2 \times N + p^3\right). \tag{24}$$

*Proof of 26.* Summing the time complexity of **A** step 24 and **C** step 23 gives a complexity in $O\left(p^2 + p^2 \times N + \mathcal{C}_{\mathrm{Lasso}}(p, N)\right)$. Since the matrix inversion need to be performed only one time for the $p$ Lasso problems at each step, the complexity of $\mathcal{C}_{\mathrm{Lasso}}(p, N)$ using 25 becomes $O(p^3 + N \times p^2)$, finally resulting in a complexity in $O\left(p^2 + p^2 \times N + p^3\right) = O\left(p^2 \times N + p^3\right) = O(\mathcal{C}_{\mathrm{Lasso}}(p, N))$. $\qquad\square$

## B  Additional experiments

### B.1  Number of iterations

For synthetic experiments in Section 4, we motivated the choice of $N_{\mathrm{iter}} = 10$ by Figure 8. While this parameter allows good results in various scenarios, it can be interesting to understand whether this parameter is related to the dataset. Convergence analysis results are shown in Figures 14 and 15. A trend seems to appear : the larger $N$, the faster the convergence. Moreover, while the noise impacts the performances of AALasso, the convergence rate seems to remain unchanged for $\sigma_A = 0.1$ or $\sigma_A = 0.25$.

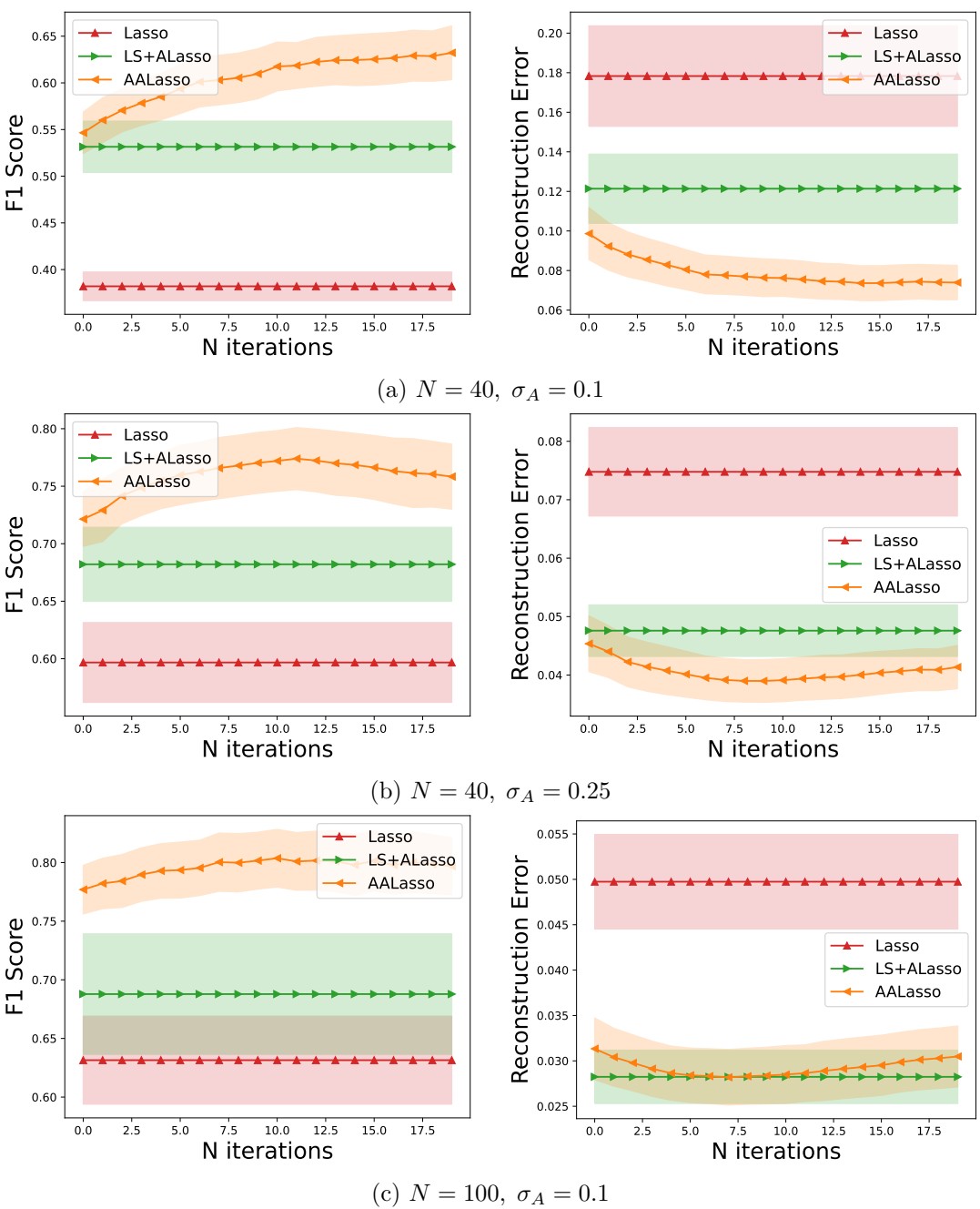

(a) $N = 40,\ \sigma_A = 0.1$

(b) $N = 40,\ \sigma_A = 0.25$

(c) $N = 100,\ \sigma_A = 0.1$

Figure 14: F1-score and Prediction error through AALasso iterations for $N \in \{40, 100, 140\}$ and $\sigma_A = 0.1$.

## B.2 Runtime

## B.3 Initialization impact

Let's recall the algorithm to train AALasso :

The results presented in the previous sections were obtained initializing $A^{(0)}$ with $\mathbf{A}^{\text{prior}}$. However, it can be interesting to test the algorithm with other initializations. We therefore conducted experiments initializing $A^{(0)}$ with only 1 values (denoted AALasso-ones), $A^{(0)}$ obtained by solving the Least Squares problem (denotes AALasso-LS) or with random values (denoted AALasso-random). Our method is proven to converge towards

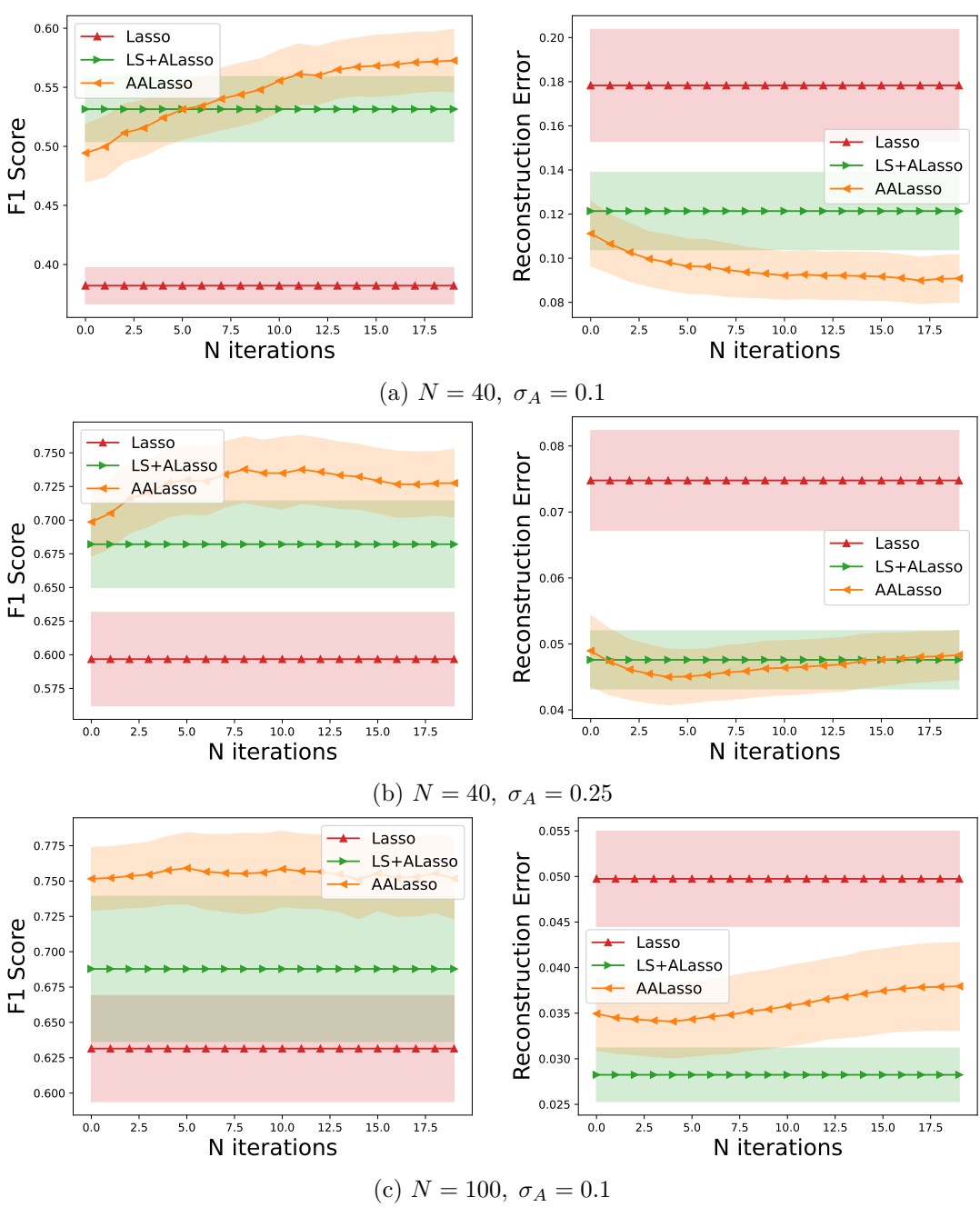

(a) $N = 40, \ \sigma_A = 0.1$

(b) $N = 40, \ \sigma_A = 0.25$

(c) $N = 100, \ \sigma_A = 0.1$

Figure 15: F1-score and Prediction error through AALasso iterations for $N \in \{40, 100, 140\}$ and $\sigma_A = 0.25$.

a stationary point, no matter the initialization of $A^{(0)}$. However, since our objective function is not convex, better local minimum could be found starting from a 1 vector or a random vector rather than the $\mathbf{A}^{\text{prior}}$. The results for all the scenarios tested are presented in Figures 16 and 17. Globally, the results are similar for all initializations tested and it is not clear if one of the initializations is better than the other. However we remark than in settings with very few samples ($N = 40$), the random initialization surprisingly outperforms the other with a gain of 2.5 in the F1-scores compared to AALasso for all noise levels.

---

**Algorithm 3:** Fitting algorithm.

---

**input** : $N_{\text{iter}}$, $\lambda$, $\gamma$, $\mathbf{A}$
**output:** $\widehat{\mathbf{C}}$, $\widehat{W}$
$W^{(0)} \leftarrow$ Subdiagonal values of $A$
**for** $i \leftarrow 1$ **to** $N_{iter}$ **do**
    $C^{(i)} \leftarrow f_{\mathbf{C}}(\mathbf{C}, W^{(i-1)})$ where $f_{\mathbf{C}}$ denotes the update in (3.4.1).
    $W^{(i)} \leftarrow f_W(\mathbf{C}^{(i)}, W)$ where $f_W$ denotes the update in (3.4.2).
**return** $\mathbf{C}^{(N_{\text{iter}})}, W^{(N_{\text{iter}})}$.

---

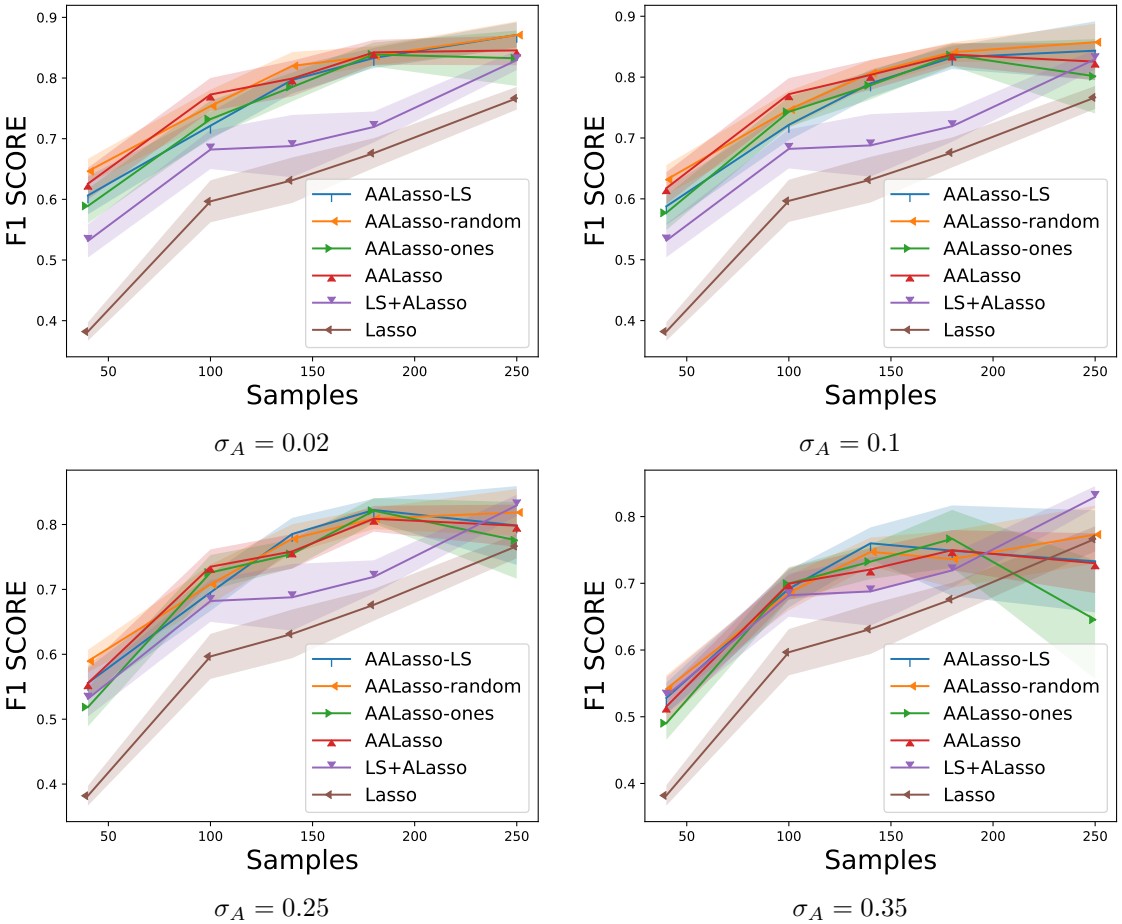

Figure 16: Impact of the initialization of $\mathbf{A}^{(0)}$ on the F1-scores with $\sigma_A \in \{0.02, 0.1, 0.25, 0.35\}$.

## B.4 Comparison with random prior graph

In this last experiment, we compare our results with the AALasso method taking a random $\mathbf{A}^{\text{prior}}$ to check that the prior structure is well leveraged and that a random L2 penalization can not achieve the same performances. The values $\mathbf{A}^{\text{prior}}_{i,j}$ are sampled independently from a uniform distribution in $[0.2, 1]$. The results are presented in Figure 18, and we see that results using a random $\mathbf{A}^{\text{prior}}$ are very poor regarding both F1-score and Prediction error. This reinforces the thesis that AALasso judiciously leverages the information provided by $\mathbf{A}^{\text{prior}}$.

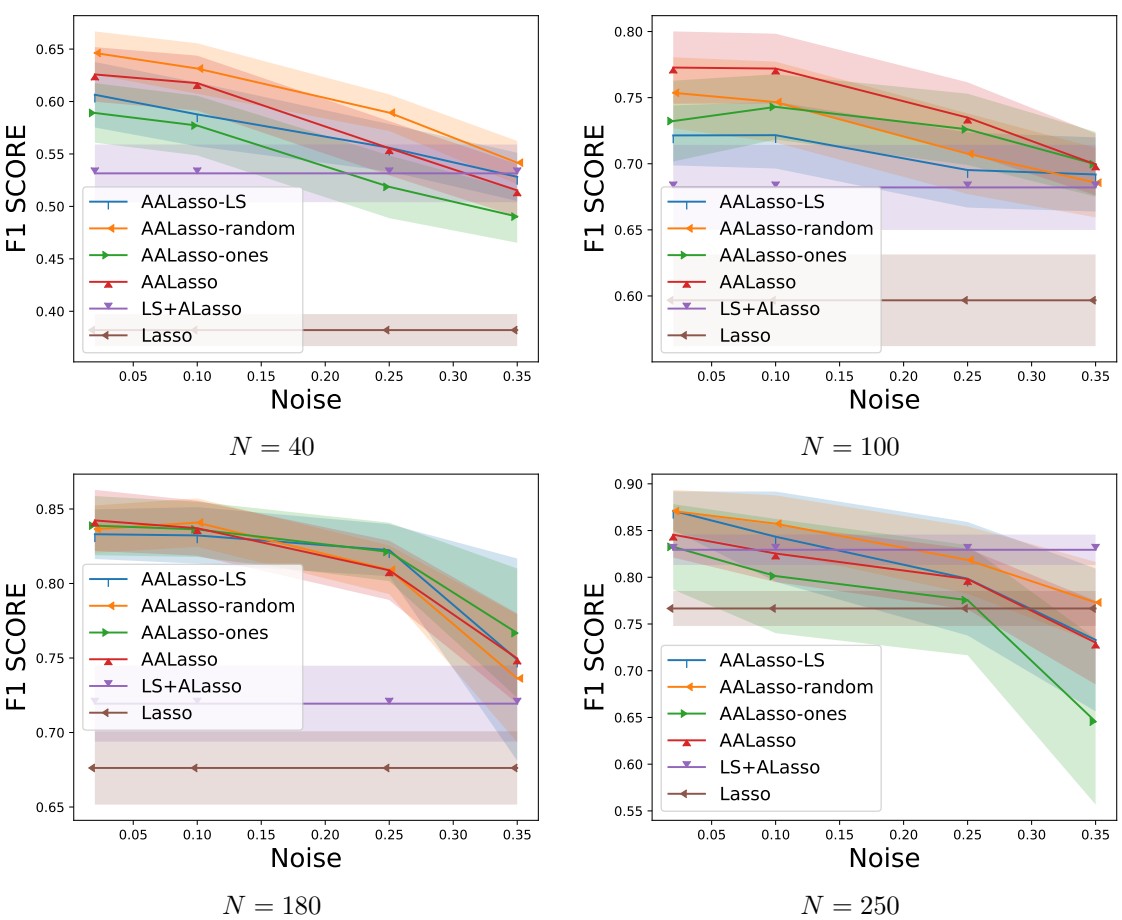

Figure 17: Impact of the initialization of $\mathbf{A}^{(0)}$ on the F1-scores with $N \in \{40, 100, 180, 250\}$.

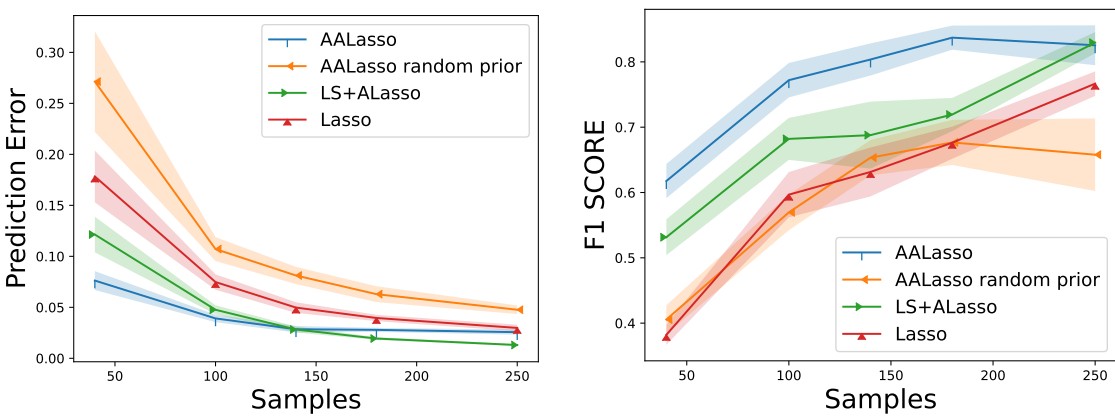

Figure 18: Comparison of AALasso, Lasso and LS+ALasso with AALasso taking $\mathbf{A}^{\text{prior}}$ random, with $\sigma_A = 0.1$.

## B.5 Runtime

Here, several scenarios are tested to complete the Figure 9 and show that the behavior regarding the F1 score in function of the runtime remains similar for some parameters choices.

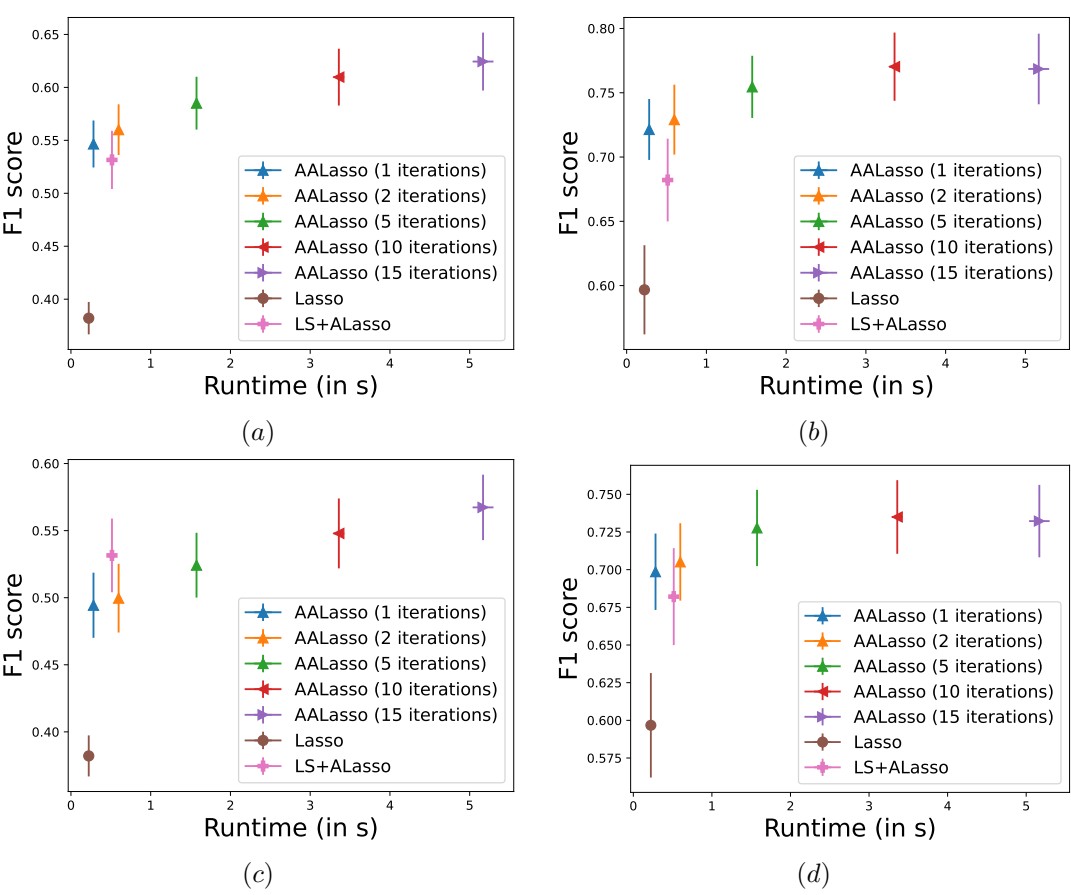

Figure 19: F1 score in function of runtimes for Lasso, LS+ALasso and AALasso for $N_{\text{iter}} \in \{0, 5, 10, 15\}$. (a) $N = 40$ samples, $\sigma_A = 0.1$. (b) $N = 100$ samples, $\sigma_A = 0.1$. (c) $N = 40$ samples, $\sigma_A = 0.25$. (d) $N = 100$ samples, $\sigma_A = 0.25$.

