# OpenReview forum: "Learning Network Granger causality using Graph Prior Knowledge"
_TMLR — Accepted by TMLR_

### Review · Reviewer_GGay · 2024-02-28

**Summary Of Contributions:**

The paper introduces a model to learn network Granger causality by using prior knowledge in the form of an undirected graph. The study aims to mitigate the challenge of estimating Granger causality graphs in high-dimensional scenarios with limited samples. Experimental results from synthetic and real-world datasets demonstrate the effectiveness of the proposed method over existing alternatives. The work has a solid theoretical and technical foundation. However, some serious issues need to be clarified and resolved. I lean toward a major revision for the work.

**Audience:**

Yes

**Broader Impact Concerns:**

No ethical concerns

**Claims And Evidence:**

Yes

**Requested Changes:**

The authors can refer to the weakness to make changes. Specifically, the authors should make the following changes.

C1: The authors should explain why non-zero matrix entry Cij can capture the Granger cause between Xi and Xj?

C2: Please add high-level intuition to illustrate the idea.

C3: Please provide a detailed explanation of how to construct the graph prior Aprior in the problem formulation.

C4: Is there a long-distance dependency between two nodes of Aprior that are far away? If yes, then the assumption may be problematic. If not, please justify the point.

C5: The authors should conduct experiments on extremely scarce data and higher dimensional data to validate performance

C6: "Equation (2) capture specific temporal dependencies between the p time series". However, p is the dimension of variables. Is this a typo?

C7: Compared to existing works, please illustrate the advantages of the proposed one and the reasons.

**Strengths And Weaknesses:**

Strength:

S1: The work has a solid theoretical foundation and support.

S2: The authors conducted extensive experiments to evaluate the performance of the proposed method.

Weakness:

W1: Although the work has a solid theoretical foundation, the work lacks high-level intuition to illustrate the idea, which establishes a barrier to a broad audience.

W2: Although the authors provided extensive preliminaries to help the reviewer delve into the research. However, some details of the network Granger causality should be clarified.

W3: It is unclear how to construct the prior matrix Aprior. The concept is suddenly introduced in problem formulation without a detailed explanation.

W4: The assumption in the problem formulation (i.e. if two nodes (i, j) are far away in the graph, then there is no Granger causality between the two-time series) seems to be problematic and needs further discussion.

W5: Compared to existing works, it is still unclear the advantages of the proposed one using the graph prior knowledge.

W6: The authors claimed the design can deal with high dimensional variables with limited samples. This case often means n<<d, where n is the number of samples and d is the dimension. However, the experiments fall short of evaluating the case.

---

### Review · Reviewer_oNFM · 2024-03-10

**Summary Of Contributions:**

The current paper proposes a method to build a vector autoregressive model for multi-variate forecasting of time series data while recovering an underlying causal network between the time series.
The problem is cast as a linear program to predict the current value of a time series by a linear combination of all the series’ history with a predefined depth.
The least squares cost function is added to a reformulation of the adaptive lasso such that the undirected edges that act as coefficients in the cost function are penalized based on a prior undirected network. The edges of the prior network are sampled uniformly from a distribution in [0.2,1]. In addition, an l2 penalty is set to adjust the prior graph according to the observation, similar to an l2 regularisation in linear regression which is known to be equivalent to setting a Gaussian prior in the learning parameters of the MLE problem. The authors develop a similar proof for the same argument on the proposed model.
The method is proved to be nonconvex and an ADMM algorithm is derived and is proved to converge in stationary points. The results indicate an improved performance in forecasting accuracy and network reconstruction error in several real-world datasets.

**Audience:**

Yes

**Broader Impact Concerns:**

There is no obvious ethical concern for the paper's broader impact.

**Claims And Evidence:**

No

**Requested Changes:**

“The closer the I and j are in the graph the lower the penalty”: My confusion here pertains to the nature of A. If A is a weighted adjacency the elements show strength, not proximity. If A indicates distance i.e. in a number of hops, it is A + A^2 + A^3 etc.. That however would mean that A is no longer sparse and the computational burden for real-world graphs would be a problem. This could be clarified further.

Some simulations indicating how the model behaves under non-stationary setting would make the paper more complete. Moreover, it is not clear if equation 16 indicates a normalization of solely the input or the whole time series. If it is the latter doesn’t it cause a change of the results?

Although explained in 3.5 and used in 4.5.2, it is not perfectly clear to me, does the model for d > 1 breaks down to d different models i.e. graphs learnt simultaneously?

Causality is inherently a directed measure so the intuition behind setting an undirected graph as a prior is not fully clear.

To verify the intuition of section 3.4, wouldn’t sampling the prior edges from a gaussian instead of uniform distribution improve the final reconstruction?

Typos: ant, wi,j, the generalize prior

**Strengths And Weaknesses:**

Strengths:
The paper is well written, with clear arguments and proofs.
The model is interpretable which is very important for practitioners in the targeted fields.
The algorithm is sound and the convergence proof for a non-convex and non-differentiable function is a useful result.
The method is tested in both simulated and real data and it improves compared to other traditional methods.
A visual explanation of the results facilitates the qualitative evaluation of the methods.

Weaknesses:
To my understanding, the method aims to reconstruct the causal network. This is useful in cases where we search for new causal relationships while observing an already established causal network i.e. enriching the network. If this is the aim of the method, it is not perfectly clear from the title or the text and the experiments.
Specifically, the simulated data utilize a noisy representation as a prior and are evaluated based on the reconstruction error with the correct network. This is not very clear with the real data, as neither the noisy network nor the reconstruction error (accuracy in 4.5.1?) is clarified sufficiently.  Similar to section 4.1. an integral experiment is to remove edges for varying ratios in the real network, use it as a prior, and calculate how close is the retrieved network with the real one.

An even more important question is the role of A in eq. 7. If C is the retrieved network and A^prior the noisy observation, what does A represent? I imagine it can not be the actual causal network that we aim to identify with C, because in the real-world this is not observed.

In addition, the method is not tested sufficiently. Comparison with other causal network discovery models https://arxiv.org/pdf/2006.10833.pdf or algorithms that do not utilize priors or facilitate prediction, such as PCMCI https://www.science.org/doi/pdf/10.1126/sciadv.aau4996 and its variants  https://openreview.net/forum?id=dYeUvLUxBQ can be beneficial to argue about the usability of the model. In accordance, comparison with other forecasting methods https://arxiv.org/pdf/1704.04110.pdf and simple average window benchmarks for the accuracy need to be added.

A final vague point pertains to the identification of confounders. The model seems to allow for causal relationships for example s1 -> s2, s1 -> s3 to create spurious edges s2 -> s3 in case s2  precedes s3 since s2 and s3 change behaviour according to s1.

---

### Review · Reviewer_Hjee · 2024-04-01

**Summary Of Contributions:**

The paper introduces a novel optimization problem that allows to estimate VAR models while incorporating the use of Granger causality graphs based on the assumption that there exists a noisy Prior causal graph of the specific multivariate times-series input. Importantly, the authors provide an efficient two-block coordinate descent algorithm while also providing guarantees for the convergence of such an algorithm. The results showcase the superiority of the method against the considered baselines. Lastly, they showcase the important of sparsity in the performance of the general model.

**Audience:**

Yes

**Broader Impact Concerns:**

No ethical concerns.

**Claims And Evidence:**

Yes

**Requested Changes:**

1) Please provide an extensive discussion on the problem setting, and on how common is the existence of a prior Granger causality information

2) The VAR coefficient matrix C as stated in the main paper expresses the directed Granger causality graph. Given that, why is A and Aprior chosen to be undirected? Would it make more sense that also the prior is directed? Please elaborate more on this modeling decision.

3) Please include more recent and complex baselines and compare how your method competes against them in discovering causality graphs.

**Strengths And Weaknesses:**

Strengths:

1) The paper provides an extensive mathematical study of the proposed two stage optimization method, and importantly convergence guarantees.

2) The loss function of equation (7) of the main paper, is well-motivated and principled under the statistical model provided in Figure (1) of the main paper.

3) The model is simple and intuitive.


Weaknesses:

1) In real-world scenarions, the problem setting where there exists even a noisy A_prior matrix is rather limiting. In most settings, such an assumption is difficult to be made. In this sense, the proposed approach is unable to work in the more general case where prior knowledge is not present.

2) The baselines are rather limited. Comparison with more recent and competitive methods that aim on learning Granger causality graphs based on time series data should be provided. I am including just a few here:

i) Amortized Causal Discovery: Learning to Infer Causal Graphs from Time-Series Data, Löwe et al, 2022

ii) Discovering Nonlinear Relations with Minimum Predictive Information Regularization, Wu et al, 2020

---

### Decision · Action_Editor_4QrC · 2024-05-22

**Recommendation:** Accept as is

**Comment:**

The reviewers provided insightful comments and the authors have responded to those comments. The clarity of presentation is much improved. One reviewer expressed some concerns about the extent of comparisons to other state-of-the-art methods. These concerns are reasonable. However, the authors scope their claim to rest upon the evidence: "Thus the incorporation of prior information has proven instrumental in achieving superior accuracy and robustness when compared to classical algorithms in this domain." Overall, the reviewers were in favor of acceptance, and I agree.

**Audience:**

Yes. There are multiple application domains that would benefit from incorporating prior network information into the causal network learning problem. While there were some weaknesses identified by reviewers, the work was judged to be a sufficient advance to warrant attention of the community.

**Claims And Evidence:**

This paper claims an approach for learning vector autoregressive (VAR) models that incorporates uncertain prior information about the structure of the model. VAR models are important to the problem of causal inference because they are used to learn Granger causal graphs. The approach is a block coordinate method and the optimization problem the approach solves is related to MAP inference for a statistical model. The paper provides evidence for the claim that the approach solves the optimization problem of interest in the form of proofs of related theorems and they show empirical evidence that the approach produces reasonable results on synthetic and biomedical data. The authors, furthermore, study the sensitivity of the model to prior specification. They conduct extensive comparisons to classical and other causal inference algorithms to show that there are meaningful benefits to incorporating both time-series and noisy prior network information in the problem of learning Granger causal networks. The claims seem to be well supported by the evidence.